# CIRBench: Evaluating Large Language Models as LLVM IR Optimizers

Zi Yang [1]   Haifeng Ding [1]   Fei Liu [1]   Yingying Cheng [1]   Han Cheng [1]   Zhilei Chai [1]   Haojie Zhou [1]

## Abstract

Large language models are beginning to introduce a new paradigm for compilation: instead of only assisting at the source level, they can operate directly on **intermediate representations (IRs)**, the compiler's internal code representation. Early studies suggest that LLM-guided optimization can sometimes rival traditional compiler optimizations on selected programs, but evidence remains fragmented. Yet the community still lacks a rigorous IR-level benchmark that tests whether a model not only understands IR but can rewrite it under compiler-grade semantic constraints with meaningful performance impact. We present **CIRBench**, a benchmark of 800 curated IR instances spanning four compiler-oriented tracks: Analysis infers IR properties, Repair fixes invalid IR, Refactor applies a single semantics-preserving compiler optimization, and Transform performs performance-oriented rewrites, together mirroring core optimization responsibilities in modern compilers. CIRBench combines verifier, equivalence checking, and end-to-end performance measurement into a unified, layered correctness-aware evaluation of LLMs on IR. On six mainstream LLMs, CIRBench shows that current models fail on many IR analysis and rewriting instances and on median underperform the compiler baseline, but we also observe a maximum speedup of $4.96\times$ over `-O3`. These findings highlight both the opportunities and the remaining challenges of using LLMs inside optimizing compilers.

## 1. Introduction

Large language models are increasingly being used not only as source-level coding assistants (Lozhkov et al., 2024; Hui et al., 2024; Roziere et al., 2023; Guo et al., 2023), but

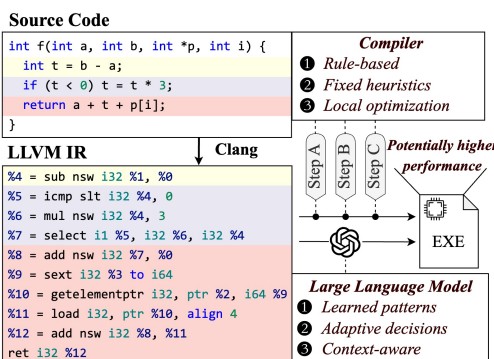

**Source Code**

```
int f(int a, int b, int *p, int i) {
    int t = b - a;
    if (t < 0) t = t * 3;
    return a + t + p[i];
}
```

**Compiler**
❶ *Rule-based*
❷ *Fixed heuristics*
❸ *Local optimization*

**LLVM IR** ← Clang

```
%4 = sub nsw i32 %1, %0
%5 = icmp slt i32 %4, 0
%6 = mul nsw i32 %4, 3
%7 = select i1 %5, i32 %6, i32 %4
%8 = add nsw i32 %7, %0
%9 = sext i32 %3 to i64
%10 = getelementptr i32, ptr %2, i64 %9
%11 = load i32, ptr %10, align 4
%12 = add nsw i32 %8, %11
ret i32 %12
```

*Potentially higher performance*

Step A / Step B / Step C → EXE

**Large Language Model**
❶ *Learned patterns*
❷ *Adaptive decisions*
❸ *Context-aware*

*Figure 1.* Comparison between traditional compiler optimization and LLM-driven compilation, showing how LLMs replace heuristics with learned patterns.

also as components *inside* compiler pipelines (Yang et al., 2026; Jin et al., 2025). Instead of writing whole functions from scratch, they are asked to inspect compiler artifacts, propose edits, and cooperate with existing optimization passes (Zheng et al., 2024; Pan et al., 2025c). This shifts the central question from "Can an LLM write a correct program?" to a stricter, deployment-oriented one: *Can a model operate on the compiler's internal representations in a way that preserves correctness under strong compiler-style semantic checks while also delivering consistent performance gains?* We study this question through a correctness-aware IR benchmark.

Modern compilers achieve portability and performance by translating many source languages into a common intermediate representation (IR) (Alfred et al., 2007)(a low-level, typed program form shared across front-ends/back-ends), running a sequence of analysis and optimization passes on this IR, and then emitting machine code. As illustrated in Figure 1, front-ends such as Clang compile C into LLVM IR (Lattner & Adve, 2004; LLVM Project, 2024a), and a rule-based optimizer applies step-by-step local transformations before producing an executable (LLVM Project, 2024c). LLM-based systems can be integrated at the same IR boundary: instead of fixed heuristics and hand-written rules (Seeker et al., 2024), the model can propose context-aware transformations while keeping the same executable interface and correctness checks.

Recent work at the intersection of compilers and machine

---

[1]School of Artificial Intelligence and Computer Science, Jiangnan University, Wuxi, China. Correspondence to: Haojie Zhou <zhouhaojie@jiangnan.edu.cn>.

*Proceedings of the $43^{rd}$ International Conference on Machine Learning*, Seoul, South Korea. PMLR 306, 2026. Copyright 2026 by the author(s).

*Table 1.* Comparison with representative IR-level resources and evaluations. A/Rp/Rf/T denote Analysis, Repair, Refactor, and Transform; V/E/P denote verifier checks, equivalence checks, and performance measurement.

| Work | IR | A | Rp | Rf | T | V | E | P |
|------|----|----|----|----|----|----|----|----|
| SLTrans / IRCoder | ✓ | – | – | – | – | – | – | – |
| ProGraML / DeepDataFlow | ✓ | ✓ | – | – | – | – | – | – |
| IR-OptSet | ✓ | ✓ | – | – | ✓ | ∼ | ∼ | ✓ |
| LLM-Vectorizer | ∼ | – | – | – | ✓ | – | ✓ | ✓ |
| CIRBench | ✓ | ✓ | ✓ | ✓ | ✓ | ✓ | ✓ | ✓ |

learning has explored a variety of IR-level applications along three axes: learning optimization decisions, synthesizing local rewrites, and prompting LLMs for end-to-end IR rewriting (Cummins et al., 2021; Dutta & Jannesari, 2024; Seeker et al., 2024). Examples include learning heuristics for inlining or register allocation (Trofin et al., 2021; Venkata-Keerthy et al., 2023), using neural or symbolic methods to synthesize local optimization rules (Taneja et al., 2025; Pan et al., 2025a), and prompting large models to rewrite unoptimized IR into more efficient forms that occasionally rival or surpass `-O3` (Wei et al., 2025). These results suggest that learned systems can discover non-obvious optimizations and act as powerful assistants to hand-engineered pipelines, rather than merely autocompleting source code.

At the same time, existing evaluations of IR-level learning are highly heterogeneous. Most works design their own tasks, or repurpose general program suites with bespoke metrics and correctness assumptions (Jiang et al., 2025; Italiano & Cummins, 2025; Zhang et al., 2025a), and, to our knowledge, there is still no single benchmark that systematically evaluates LLMs on compiler IR across both analysis and semantics-preserving optimization with layered correctness and performance metrics. This makes side-by-side comparison difficult and obscures where models are reliably helpful versus risky, motivating a shared, layered correctness-aware benchmark that lives directly at the IR level, where small semantic slips can silently miscompile programs. Table 1 summarizes this gap: prior IR-level resources typically cover only part of analysis, rewriting, semantic checking, or performance measurement, whereas CIRBench evaluates Analysis, Repair, Refactor, and Transform under a shared LLVM IR interface and correctness-aware protocol.

In this paper we introduce **CIRBench**, a layered correctness-aware benchmark for evaluating LLMs on LLVM IR. CIRBench is built around four compiler-oriented tracks that jointly probe IR *understanding* and *rewriting*: (i) *Analysis* asks whether a model can infer properties that compilers rely on, such as alias information and loop structure; (ii) *Repair* asks a model to fix IR that fails verification; (iii) *Refactor* measures a model's ability to implement a *given* optimiza-

tion pass; and (iv) *Transform* evaluates more open-ended IR optimizations that aim to achieve better performance than compiler `-O3`. To mirror compiler practice, CIRBench uses a layered best-available semantic validation protocol rather than a single homogeneous proof notion: candidate IR must first pass the LLVM verifier (LLVM Project, 2024a); verifier-clean single-function cases are checked with Alive2 (Lopes et al., 2021) when it returns a definite verdict; otherwise, and for module-level cases outside Alive2's scope, we use checksum-based execution over deterministic harness inputs (Maleki et al., 2011). We never override an Alive2 NOT-EQ verdict with harness agreement. On top of these checks we report Valid@k and Equiv@k for $k \in \{1, 5\}$ on generation tracks, exact match(EM) and F1 on Analysis-style prediction tracks, and then performance relative to `-O3` measured via both static cost models and real execution (details in Sections 3 and 4) (LLVM Project, 2024b).

We apply CIRBench to six mainstream LLMs (OpenAI, 2025a; Anthropic, 2025; Google AI / DeepMind, 2025; xAI, 2025; Qwen Team, 2025; DeepSeek AI, 2025), using the LLVM toolchain as a reference baseline (LLVM Project, 2024d). Models can already solve a subset of Analysis and Repair cases, but remain fragile on control-flow restructuring, symbol resolution, and fine-grained attribute reasoning (Jiang et al., 2025). On the Transform track, strict semantic guards reveal a gap between syntactic validity and semantic equivalence: the best model reaches up to $4.96\times$ speedup over `-O3` on individual kernels, yet its median dynamic speedup is only $0.85\times$, i.e., slower than the compiler baseline. Strengthening IR understanding and optimization is therefore a long-term goal; we later distill several observations and design hints from CIRBench for both general-purpose LLMs and specialized IR models.

Our contributions are three-fold:

- We present **CIRBench**, a layered correctness-aware benchmark for LLVM IR that organizes IR reasoning into four tracks—Analysis, Repair, Refactor, and Transform—each tied to explicit semantic oracles.

- We define a standardized evaluation protocol and open-source toolkit that reflect compiler practice, with layered correctness metrics and fully reproducible pipelines.

- We provide an empirical study of six mainstream LLMs on CIRBench, establishing baseline results, revealing concrete failure modes, and quantifying the headroom and offering actionable directions for future IR-level learning methods.

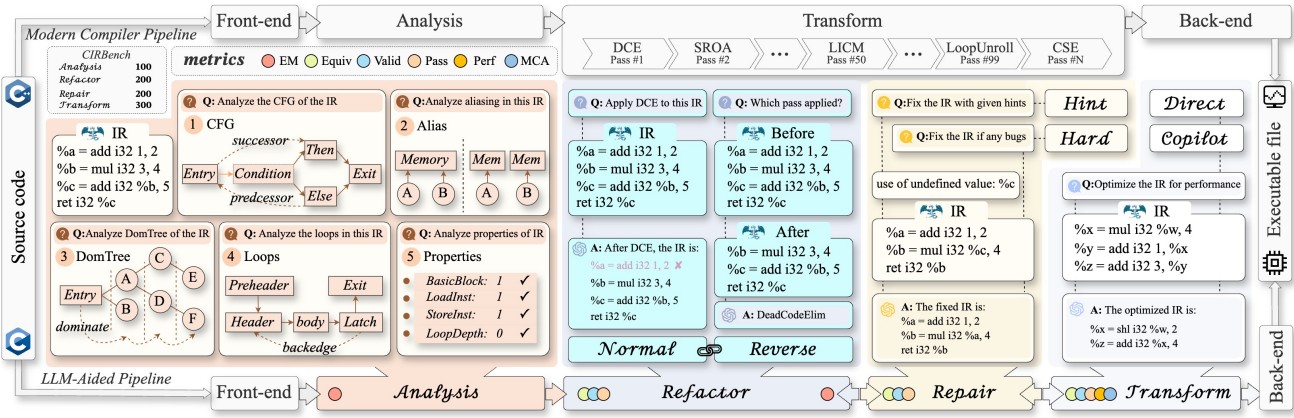

*Figure 2.* Overview of CIRBench within a conventional compiler pipeline. The upper row shows the usual front-end–middle-end–back-end flow; the lower row replaces parts of the middle-end with LLM-based Analysis, Refactor, Repair, and Transform tasks, while keeping the same IR interface and executable output.

## 2. Background

### 2.1. Intermediate Representations and Correctness

Modern compilers translate many source languages into a common *intermediate representation* (IR) (Alfred et al., 2007) and apply a series of optimizations on IR. LLVM IR is a widely used example: a strongly typed, SSA-based IR with explicit basic blocks and instruction-level attributes (LLVM Project, 2024a). Most performance-critical optimizations are implemented as passes that rewrite IR while preserving program behavior (LLVM Project, 2024c).

What matters for this paper is not the full pipeline, but that IR semantics are both expressive and brittle. LLVM IR exposes behaviors that high-level languages mostly hide: undefined behavior (UB), *poison* values, and attributes such as nuw (integer overflow flags) and readonly (function properties) (LLVM Project, 2024a). For example, a single incorrect nuw flag can change overflow semantics. These features enable aggressive optimizations, yet a single incorrect flag, missing store, or change in dominance can silently turn a valid program into a miscompilation (Kasampalis et al., 2021; Lopes et al., 2021). Thus any learning system that edits IR must be evaluated under layered semantic correctness constraints before performance is considered.

### 2.2. Learning at the IR Level

Learning-based models trained on code and natural language, including LLMs, are increasingly being applied directly to compiler IR (Gong et al., 2025). Typical tasks include: (i) producing high-level descriptions or summaries of IR fragments (VenkataKeerthy et al., 2020; Dutta & Jannesari, 2024; Zhang et al., 2022); (ii) predicting results of compiler analyses such as vectorizability or aliasing (Taneja et al., 2025); (iii) repairing IR that fails verification (Lopes et al., 2021; Zhang et al., 2025b); and (iv) rewriting un-optimized IR (e.g., near -O0) into a form closer to what -O3 produces, sometimes with the goal of surpassing it on specific kernels (Yang et al., 2026).

Earlier work mostly learned *policies* around an otherwise fixed pipeline—for example, inlining heuristics or pass ordering (Trofin et al., 2021; Pan et al., 2025b)—so correctness was inherited from existing passes. More recent IR-level uses intervene on the IR itself, placing correctness back onto the learned system, closer in spirit to superoptimization and equality saturation (Fan & Regehr, 2024; Sasnauskas et al., 2017), but typically at larger granularity and without a built-in proof procedure.

In this setting, IR optimization naturally becomes the focus of learning. Conceptually we can distinguish three layers: *Analysis*, which infers properties of IR programs; *Refactor*, which normalizes the shape of IR (control flow, SSA form, attributes) without changing semantics; and *Transform*, which performs semantics-preserving rewrites under a cost model to improve performance or code size (Grubisic et al., 2024). *Repair* sits between *Analysis* and *Refactor*, focusing on restoring broken IR to a safe, well-formed state. Compared with source-level benchmarks, IR exposes control and data flow and optimization-relevant attributes explicitly, making it a natural target for compiler-oriented LLMs. This decomposition directly motivates the four tracks in CIRBench.

## 3. CIRBench

### 3.1. Overview

Figure 2 shows how CIRBench is positioned relative to a conventional compiler pipeline. The upper path is the standard "front-end–middle-end–back-end" flow: source code is compiled to LLVM IR, a sequence of rule-based passes

rewrites the IR, and the result is lowered to an executable. The lower path reuses exactly the same interface, but allows an LLM take over part of the middle-end: the IR produced by the front-end can be sent either to classic passes or to a model that answers analysis questions, repairs invalid IR, performs single-pass refactorings, or proposes more aggressive IR-to-IR transforms, and the resulting IR is still handed to the back-end to produce runnable code. In other words, CIRBench exercises models at the same IR boundary where production compilers already make optimization decisions, rather than on an abstract toy representation. All four tracks share a consistent LLVM IR interface and jointly mirror real middle-end responsibilities.

*Figure 3.* Distribution of Easy/Medium/Hard instances across the four tracks in CIRBench.

In total, CIRBench contains 100 Analysis, 200 Refactor, 200 Repair, and 300 Transform instances. All are extracted from LLVM IR of real programs that compile to executables; Analysis/Refactor/Repair use function- or module-level slices, while Transform instances come with a harness and compile to standalone binaries. Within each track, we stratify instances into Easy/Medium/Hard levels according to IR size and structural complexity (e.g., CFG shape and data-flow depth). Concretely, difficulty is based on instruction/basic-block counts and control/data-flow complexity (e.g., loop nesting, number of backedges, and maximum def–use depth), with track-specific thresholds in Appendix G. Figure 3 illustrates the distribution of instances across the four tracks. Prompt templates and example question–answer pairs for all tracks are provided in Appendix B.

### 3.2. Tracks

**Analysis.** In compilers, static analyses (CFG, dominators, loops, aliasing, etc.) are the precondition for safe and profitable optimizations: if an analysis is wrong, the downstream transform either misses an opportunity or miscompiles (TehraniJamsaz et al., 2023). The Analysis track asks whether a model can recover such analysis facts directly from IR.

We collect 100 LLVM IR functions, grouped into five categories, with 20 functions per category, and design short analysis questions as illustrated in the left panel of Figure 2. Typical prompts ask, for example, "What is the CFG of this function?" and expect a JSON answer that, for each basic block, lists its predecessor blocks. Ground-truth JSON is obtained from LLVM's built-in analyses and auxiliary scripts, so that it matches what a production compiler would

compute.

*Table 2.* Five categories of IR analysis subtasks in CIRBench. Prevalence indicates the frequency of these analyses in the standard -O3 pipeline.

| Category | Prevalence (%) | Easy / Medium / Hard | | |
|---|---|---|---|---|
| A1: Alias | **17.1** | 8 | 7 | 5 |
| A2: Loops | **29.8** | 6 | 6 | 8 |
| A3: Dominator Tree | **51.7** | 7 | 7 | 6 |
| A4: CFG | **28.8** | 5 | 7 | 8 |
| A5: Basic Properties | - | 9 | 7 | 4 |

We group questions into five subtasks in Table 2: A1 (ALIAS) labels pointer pairs with {NoAlias, MayAlias, MustAlias, PartialAlias}; A2 (LOOPS) recovers loop headers and nesting; A3 (DOMTREE) asks which blocks dominate which; A4 (CFG) asks for predecessors; and A5 (PROPERTIES) requests simple statistics such as the number of basic blocks, loads, and stores. The "Prevalence" column reports how often each category is used in LLVM's transformation pipeline, and the Easy/Medium/Hard split reflects IR size and structural complexity. Prevalence is measured on LLVM's default -O3 pass pipeline (details in Appendix C).

At evaluation time we parse model outputs as JSON and compute EM and F1 against oracle answers; malformed JSON is counted separately. We canonicalize JSON before computing EM; additional metrics and formats are detailed in Appendix C. This turns Analysis into a clean prediction task over IR, with a single scalar EM score per model. Representative prompts and JSON formats are given in Appendix B.

**Refactor.** Refactor tests whether a model can perform *individual* LLVM optimization passes (rather than imitating the net effect of -O3). Each instance targets one pass (e.g., DCE, SROA, LICM, Loop Unrolling), following Figure 2. We choose 20 frequent passes according to IR-Optset (Yang et al., 2026) and build 10 instances per pass (200 total), split into Easy/Medium/Hard based on IR size and control-flow complexity.

We consider two modes.

*Refactor-Normal.* Given an unoptimized IR fragment (typically a full function, including any required surrounding module context)and a single pass name (e.g., Apply DCE), the model must apply *only* that pass and return the optimized IR. For up to $k$ samples $s_1, \ldots, s_k$, we define

$$\textbf{Valid}@k = \mathbf{1}\big[\exists j \le k : \mathrm{Ver}(s_j) = \mathrm{ok}\big],$$

$$\textbf{Equiv}@k = \mathbf{1}\big[\exists j \le k : \mathrm{Ver}(s_j) = \mathrm{ok} \wedge \mathrm{Eq}(s_j, \mathrm{ref})\big]$$

where Ver is the LLVM verifier, Eq is our semantic-equivalence predicate, and ref is the reference IR. For

single-function tracks, $\text{Eq}(s, \text{ref})$ uses Alive2 when it returns a definite result. When Alive2 is inconclusive or tool-limited (e.g., timeout or out-of-memory), or for module-level tasks where Alive2 is not applicable, we fall back to checksum-based harness tests over observable outputs (subsection F.8). This fallback is checksum-based rather than fully formal. We use Equiv@$k$ as our main semantic guard, while EM@$k$ (exact match after normalization) serves as a stricter, syntax-level metric. Refactor-Normal metrics are Valid@1/5, Equiv@1/5, and EM@1/5.

*Refactor-Reverse.* Given a before/after IR pair, the model identifies which of the 20 passes produced the transformation; we score by exact match. Full pass list and construction details are in Appendix D.

**Repair.** Repair evaluates fixing IR that violates structural invariants (SSA well-formedness, type consistency, meta-data, etc.) (Jiang et al., 2025). Such errors are often subtle; e.g., vector `shufflevector` masks are easy to describe but brittle to modify, where changing one index may silently alter semantics.

We inject 200 verifier-clean functions with 21 structured error types from common LLVM bug patterns and prior studies (Yang et al., 2026). Difficulty varies by function length and by injecting 1–3 errors.

Two modes follow Figure 2. In *Repair-Hint* mode, the prompt includes the verifier's error message and location. In *Repair-Hard* mode, the model only sees raw IR and must detect invalidity before repairing. Outputs are complete IR fragments. Each corrupted instance is derived from a verifier-clean reference function, so there is always at least one semantics-preserving repair (the original IR), and we treat any IR equivalent to this reference as a valid fix (Appendix E).

Repair uses the same Valid@k and Equiv@k as Refactor; we report Valid@1/5 and Equiv@1/5 to separate merely passing verification from restoring semantics. Error taxonomy and more examples are in Appendix E.

**Transform.** Transform is the most realistic and demanding track in CIRBench, closest to compiler practice. The model acts as an IR-level optimizer: given an unoptimized kernel, it may apply any semantics-preserving IR-to-IR rewrite to improve performance. Instances can be function-scoped or module-scoped; in function mode we ask the model to rewrite and return only the target function body while preserving the rest of the module unchanged.

We build 300 instances, organized into five sets of 60: (i) loop-centric kernels, (ii) core ML/DL operators, (iii) classic algorithmic/data-structure kernels, (iv) small module-level kernels with multiple functions, and (v) a *Challenge* set of high-headroom kernels curated using autotuning, PGO, and performance profiling. The first three sets reflect typical middle-end workloads; the module-level set goes beyond single functions; the Challenge set serves as a stress-test for difficult but practically meaningful patterns and is not intended to reflect prevalence in general workloads.

Each instance provides unoptimized IR and a requirement to improve run time without changing semantics; within scope the model may propose any rewrite. Our reference implementation evaluates on an x86-64 Zen4 back-end with fixed compiler flags, but the tasks are defined at the IR level and can be retargeted to other back-ends (Appendix F).

We evaluate in two modes (Figure 2): *Transform-Direct* checks and measures the candidate IR without further optimization, while *Transform-Copilot* first runs the candidate through the standard `-O3` pipeline, emulating an "LLM co-pilot" that augments `-O3` and testing whether models help more cooperatively or standalone. In Copilot, the LLM-produced IR is used as the starting IR for `opt -O3`, and we measure the IR after `-O3`.

For verifier- and equivalence-clean candidates, we measure (i) static performance via `llvm-mca` and (ii) dynamic performance by compiling and timing binaries under fixed hardware/flags, reporting speedup over `-O3` for both. All kernels share a common harness for fair per-model comparison. Appendix F details instance groups and performance profiles.

### 3.3. Benchmark Construction and Toolchain

For Analysis, Refactor, Repair and Transform, we begin from established suites such as IR-Optset (Yang et al., 2026), the LLVM test suite (LLVM Project, 2024e), LFK (McMahon, 1986), TSVC (Taneja et al., 2025), PolyBench (Pouchet et al., 2016), and a small set of hand-written core operators. These programs are compiled to LLVM IR, sliced into function-level (and a few module-level) fragments, normalized, and annotated with oracle labels and performance measurements by a custom toolchain. We then manually curate a diverse but compact subset, filtering out examples that are trivial, dominated by undefined behavior, or overly tied to platform-specific micro-optimizations.

**Protocol summary.** We compile sources with clang/LLVM (Appendix H) to obtain unoptimized IR, slice primarily at function level (with a small module-level subset preserving in-scope callees), normalize IR by renaming SSA values and stripping non-semantic metadata, and compute oracles via LLVM analyses, Alive2, and harness runs, with additional filtering for near-duplicates, UB-dominated fragments, and unstable timing behavior. Full thresholds, scripts, and toolchain details are provided in Appendix H.

We do not attempt to guarantee the absence of pretraining contamination; CIRBench is derived from long-standing public suites and transformed into normalized and corrupted LLVM IR. Therefore, CIRBench should be interpreted as a reliability benchmark at the LLVM IR boundary rather than as a contamination-free reasoning benchmark. Appendix I discusses data sources, potential overlap with common training corpora, and limitations in more detail.

The Repair track is constructed by injecting structured errors into verifier-clean IR from the same pools; for each instance we ensure that at least one reasonable repair exists that restores verifier cleanliness while preserving intended behavior.

The same toolchain is reused unchanged during evaluation, minimizing train–test mismatch and simplifying reproduction and extension. We report mean metrics with 95% bootstrap confidence intervals over instances (Appendix L), and provide per-track prompts, examples, and breakdowns in Appendix B–Appendix F.

## 4. Experiments

We focus on six mainstream LLMs on CIRBench across all four tracks: GPT-5, Claude Sonnet 4.5, Gemini 2.5 Flash, Grok 4 Fast, Qwen3-Max, and Deepseek-V3.2-Exp. In addition, we also report results for three smaller open-weight models in Appendix J.

### 4.1. Experimental Setup

**Prompting and decoding.** All tracks use the fixed prompt templates in Appendix B. For generation-style tracks (Refactor-Normal, Repair, Transform), we sample $k \in \{1, 5\}$ candidates per instance with identical decoding budgets across models. We use nucleus sampling with temperature and top-$p$ fixed per model family, and cap generations by a uniform stop rules at function/module boundaries (full hyperparameters in Appendix K). The same prompting and decoding protocol also applies to the three additional open-weight models in Appendix J.

**Layered correctness evaluation.** For every generated candidate we apply the same layered protocol: (i) the LLVM verifier is a syntactic gate (Valid); (ii) for verifier-clean candidates we check equivalence with Alive2 when it returns a definite result; otherwise we use checksum-based harness tests (Equiv); (iii) performance is measured only for Equiv-clean candidates. We report Valid@$\{1, 5\}$, Equiv@$\{1, 5\}$, and EM/F1 where applicable. We emphasize that Equiv@k is a best-available semantic guard rather than a uniform formal-proof metric. In our current construction, Repair and Refactor are function-level and are checked with Alive2 whenever candidates pass verification. Transform contains

both function-level and module-level cases; module-level cases, and function-level cases where Alive2 is inconclusive or tool-limited, use checksum-based harness validation. Thus, checksum agreement should be interpreted as test-based evidence of equivalence, not as a formal proof.

**Toolchain and hardware.** We use clang/opt/llc from LLVM version 19.1.0 with fixed flags described in Appendix H. Our main results report Transform dynamic speedups on an AMD Ryzen 9 7950X (Zen 4) CPU with pinned cores and a shared harness; each binary is timed for 10 runs and averaged (see Appendix F). To check that our conclusions are not specific to a single microarchitecture, Appendix Q repeats the Transform evaluation on an additional CPU back-end under matched compilation flags and harness settings.

**Confidence intervals.** All reported EM, Valid@k, Equiv@k, and speedup statistics include 95% bootstrap confidence intervals over instances(details in Appendix L).

### 4.2. Analysis Track

*Table 3.* EM and F1 accuracy on five IR analysis subtasks. Cell background color indicates difficulty: Easy/Medium/Hard (□/□/□).

| Model | EM(%) | F1(%) | A1 | A2 | A3 | A4 | A5 |
|---|---|---|---|---|---|---|---|
| GPT | 70 | 84.0 | 4 1 5 | 5 6 2 | 7 7 1 | 4 6 7 | 9 5 1 |
| Claude | 59 | 75.3 | 3 1 3 | 3 4 3 | 7 7 2 | 3 5 4 | 8 5 1 |
| Gemini | 52 | 74.0 | 3 2 1 | 4 4 2 | 7 5 1 | 2 2 5 | 8 5 1 |
| Grok | 68 | 82.0 | 4 3 5 | 3 4 2 | 7 7 1 | 5 6 5 | 9 5 2 |
| Qwen | 52 | 72.2 | 4 1 3 | 3 5 2 | 4 2 | 4 7 3 | 8 5 1 |
| Deepseek | 41 | 66.4 | 3 1 3 | 3 4 1 | 3 3 | 5 7 1 | 4 2 1 |

**Overall performance.** Table 3 reports canonicalized EM and F1 on the five Analysis subtasks. EM spans a wide range across model families (from the low 41% to around 70%), showing that IR understanding remains a clear bottleneck even before any rewriting is attempted. Errors concentrate on long IRs and deep/nested control flow, consistent with the brittleness of long-range reasoning without explicit graph structure.

**Subtask breakdown.** *A1 (Alias)* is the most difficult. Even the strongest model only solves about half of Hard instances, and smaller models fall to near-chance performance. This gap has direct downstream implications: alias imprecision can either block profitable transforms or, worse, encourage unsafe rewrites.

*A2–A4 (Loops, DomTree, CFG)* show a similar pattern: models handle shallow graphs, but degrade sharply on Hard cases. Typical failures include incomplete CFGs (missing branches), incorrect dominance relations (local correct edges but wrong transitive structure), and mis-identified

loop nesting. These results echo earlier findings that control/data-flow reasoning without structured IR views remain fragile.

*A5 (Basic properties)* is close to saturated on Easy/Medium. Yet even here, rare intrinsics or overlooked loads/stores correlate with later Repair/Transform errors, suggesting that "minor counting slips" can snowball into semantic bugs.

**Takeaway.** CIRBench shows that coarse IR structure is increasingly within reach, but fine-grained memory and loop reasoning are not. This distinction is important because many downstream IR rewrites depend less on recognizing local syntax than on maintaining a globally consistent view of data flow, control flow, and memory effects. Practically, IR-centric LLM systems should either (i) augment LLMs with graph/alias specialists, or (ii) treat uncertain analysis results as soft signals rather than hard gates. We therefore view Analysis as a necessary foundation but not a solved prerequisite: progress on IR optimization will likely require models that can expose, verify, or revise their intermediate analysis facts rather than directly emitting rewritten IR.

### 4.3. Repair Track

**Overall correctness.** Across models, Valid@k is substantially higher than Equiv@k (Table 4), indicating that surface LLVM syntax/typing is learned more reliably than the deeper semantic invariants that survive equivalence checking.

**Sampling helps, but semantics remain brittle.** Equiv@5 consistently exceeds Equiv@1 by several points, showing that multiple samples explore different repair sites or strategies. However, many candidates that pass verification still fail equivalence: repairs are not deterministic once bugs are localized, because IR semantics depend on subtle UB/attribute/calling-convention constraints.

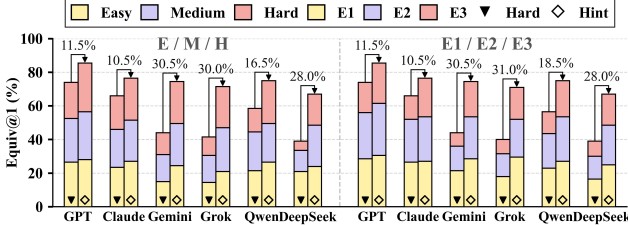

*Figure 4.* Equiv@1 on the Repair track for six models, stratified by difficulty level (Easy/Medium/Hard) and error count. (E1/E2/E3)

**Difficulty structure.** Figure 4 decomposes Equiv@1 by E/M/H and by injected error count (E1/E2/E3). Two stable trends emerge: (i) longer/more complex IRs and multi-error cases degrade Repair sharply in Hard mode; models often fix the first visible issue and stop reading. (ii) verifier hints

mitigate this length sensitivity, improving Medium/Hard and multi-error cases by 10–30% for strong models. Compiler-style diagnostics therefore act as high-value supervision.

**Takeaway.** With hints, LLMs are already useful IR repair assistants for many cases. Repair difficulty increases sharply with IR size and the number of injected errors, making long multi-error instances substantially harder than short single-error cases. The results also suggest a promising human/compiler-in-the-loop use case: compiler diagnostics can localize likely failure sites, while the model proposes candidate fixes that are then filtered by verifier and equivalence checks. But the persistent Valid–Equiv gap shows that "passes verifier" is not a sufficient correctness condition. Any deployment should keep equivalence guards in the loop and budget for multiple samples when hints are absent.

### 4.4. Refactor Track

**Refactor-Normal: executing a named pass is hard.** Refactor-Normal isolates pass-level rewriting. As Table 4 shows, Equiv@1 remains low for all models, and EM@k for the exact-pass oracle is near-zero. The main failure mode is not local arithmetic, but IR-wide invariants: metadata, attributes, globals, and intrinsics. A common failure mode is that, when the model output passes the LLVM verifier, it may still deviate from the intended single-pass behavior.

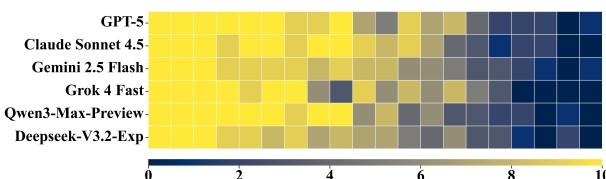

*Figure 5.* EM accuracy on the Refactor-Reverse closed-set pass-identification task across 20 LLVM optimization passes. Low scores on overlapping algebraic cleanups should be interpreted as pass-identity confusion, not necessarily semantic invalidity.

**Refactor-Reverse: recognizing passes is easier than applying them.** Refactor-Reverse is a closed-set pass-identification task over the 20 candidate LLVM passes, so exact match measures recovery of pass identity rather than general semantic plausibility. Some passes can have overlapping visible effects, such as GVN and InstSimplify; in such cases, EM is intentionally conservative and may count pass-ambiguous but semantically plausible explanations as errors. Figure 5 summarizes Refactor-Reverse across 20 passes. Patterns with visually salient, rare signatures (e.g., `GlobalOpt`, `SROA`, `TailCallElim`) are recognized at high accuracy, while subtle algebraic cleanups with overlapping effects (e.g., `GVN`, `InstSimplify`) remain confusing. This gap suggests that models have partial pass literacy but lack the precision needed for controllable pass execution.

*Table 4.* Valid@{1,5}, Equiv@{1,5} on the Repair, Refactor, and Transform tracks. We additionally report EM@1,5 for the Refactor track.

| Model | Repair-Hint | | | | | | | | Refactor-Normal | | | | | | Transform | | | |
|---|---|---|---|---|---|---|---|---|---|---|---|---|---|---|---|---|---|---|
| | Valid@ | | Equiv@ | | Valid@ | | Equiv@ | | Valid@ | | Equiv@ | | EM@ | | Valid@ | | Equiv@ | |
| | 1 | 5 | 1 | 5 | 1 | 5 | 1 | 5 | 1 | 5 | 1 | 5 | 1 | 5 | 1 | 5 | 1 | 5 |
| GPT-5 | **94.0** | **98.5** | **85.5** | **92.0** | **79.5** | **91.0** | **74.0** | **87.0** | 42.0 | 57.0 | 22.5 | 29.5 | 0.0 | 0.0 | **80.6** | **97.0** | **73.3** | **92.7** |
| Claude Sonnet 4.5 | 91.0 | 97.5 | 76.5 | 83.0 | 75.5 | 88.5 | 66.0 | 80.5 | **48.0** | **65.0** | **24.0** | **31.0** | 0.0 | 0.0 | 38.3 | 65.0 | 28.0 | 50.0 |
| Gemini 2.5 Flash | 83.0 | 91.0 | 74.5 | 83.0 | 50.0 | 74.5 | 44.0 | 66.0 | 36.5 | 62.5 | 15.5 | 28.5 | 0.0 | 0.0 | 71.6 | 93.8 | 31.5 | 46.6 |
| Grok 4 Fast | 76.5 | 94.0 | 71.5 | 89.5 | 48.5 | 77.0 | 41.5 | 72.0 | 33.5 | 56.5 | 16.5 | 27.5 | 0.0 | **0.1** | 41.7 | 85.7 | 31.7 | 76.3 |
| Qwen3-Max-Preview | 83.5 | 96.0 | 75.0 | 87.0 | 62.0 | 85.5 | 56.5 | 80.5 | 24.0 | 37.0 | 10.5 | 18.0 | 0.0 | **0.1** | 56.3 | 90.6 | 37.3 | 75.3 |
| Deepseek-V3.2-Exp | 77.5 | 91.0 | 67.0 | 79.5 | 48.5 | 73.0 | 39.0 | 61.5 | 27.5 | 57.0 | 11.5 | 24.0 | 0.0 | 0.0 | 56.3 | 91.3 | 15.3 | 40.7 |

**Takeaway.** Refactor is not subsumed by Transform: a system that occasionally beats -O3 but cannot reproduce or recognize individual passes is hard to control, debug, or integrate into production pipelines. The gap between pass recognition and pass execution also shows that current models have partial compiler literacy: they often recognize characteristic rewrite patterns, but do not yet reliably preserve the exact invariants required by a named pass. These results motivate structured IR representations, pass-aware training data, and evaluation settings that distinguish open-ended optimization from controllable compiler-pass behavior.

### 4.5. Transform Track

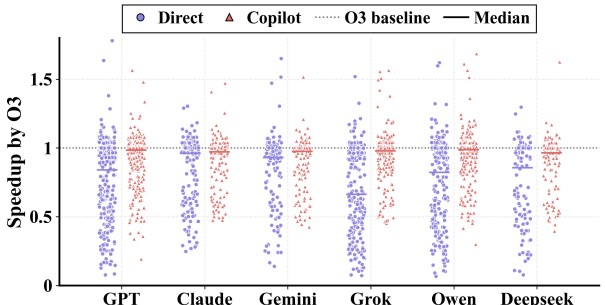

*Figure 6.* Distribution of dynamic speedup relative to -O3 on the Transform track. The dashed line at 1.0 represents parity with the -O3 baseline; values above 1.0 indicate improvement.

**Correctness and failure modes.** Transform is the most demanding track. Valid@1 is moderately high for strong models, but Equiv@1 is consistently lower (Table 4), showing that correctness failures are dominated by semantic rather than purely syntactic errors. Notably, GPT-5 breaks this trend, achieving an Equiv@5 of 92.7%, which surpasses all other evaluated models by 15–50%. Table 6 reports candidate-level failure counts over the k=5 Transform generation pool. The categories are overlapping: one candidate can simultaneously have type/operand errors, broken control flow, and symbol-resolution failures. Outputs that fail parsing or verification do not enter semantic-equivalence checking, so Semantic Mismatch is reported only for candidates that pass the structural gate but fail equivalence.

**Performance under correctness.** Figure 6 visualizes speedups over -O3. Most candidates skew towards slowdowns ($< 1.0\times$); consistent with this, in Direct mode every model's median dynamic speedup is below parity (Table 5). Thus, even when equivalence holds, more than half of candidates slow down relative to the compiler. Copilot mode (measuring post--O3 IR) truncates the negative tail and moves medians toward parity, but also dampens some extreme superoptimizing wins.

**Takeaway.** LLMs can occasionally discover high-quality IR rewrites that outperform -O3 under layered guards, but these successes are sparse and mixed with many neutral or negative cases. This makes Transform less like ordinary code generation and more like speculative optimization: the model can be valuable as a generator of candidate rewrites, but only when paired with strong filters for semantic safety and performance benefit. Practical LLM-aided compiler systems therefore need predictors of when a rewrite is likely beneficial, or cooperative settings such as Copilot mode that hedge risk by letting classical passes sanitize outputs. The positive tail nevertheless indicates real headroom: even when median performance remains below the compiler baseline, model-generated rewrites sometimes expose optimization opportunities that are not captured by the default pipeline.

### 4.6. Summary

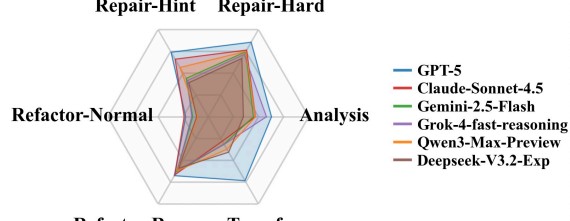

*Figure 7.* Overall performance of six models across all tracks.

*Table 5.* Performance of six models on the Transform task under direct and copilot modes, reported as speedup relative to the -O3 baseline. ↑/↓ indicate above/below 1.0×.

| Model | direct | | | | | | copilot | | | | | |
|---|---|---|---|---|---|---|---|---|---|---|---|---|
| | llvm-mca | | | perf | | | llvm-mca | | | perf | | |
| | min | med | max | min | med | max | min | med | max | min | med | max |
| **GPT-5** | 0.21 | 0.93↓ | **6.41** | 0.08 | 0.85↓ | 4.89 | 0.53 | 1.17↑ | **8.56** | 0.19 | **0.99**↓ | 3.23 |
| **Claude Sonnet 4.5** | 0.21 | 0.95↓ | 4.43 | **0.25** | **0.96**↓ | 4.53 | 0.87 | 1.21↑ | 8.11 | **0.47** | 0.98↓ | 2.36 |
| **Gemini 2.5 Flash** | 0.21 | **1.00**↑ | 4.07 | 0.14 | 0.86↓ | 4.64 | **0.97** | **1.23**↑ | 4.38 | 0.42 | 0.92↓ | 4.50 |
| **Grok 4 Fast** | **0.22** | 0.87↓ | 6.18 | 0.07 | 0.66↓ | 4.09 | 0.54 | 1.17↑ | 6.36 | 0.45 | 0.98↓ | **4.64** |
| **Qwen3-Max-Preview** | 0.21 | 0.90↓ | 4.44 | 0.07 | 0.83↓ | 4.89 | 0.56 | 1.17↑ | 4.07 | 0.30 | **0.99**↓ | 1.69 |
| **Deepseek-V3.2-Exp** | **0.22** | 0.94↓ | 4.43 | 0.08 | 0.90↓ | **4.96** | 0.81 | 1.22↑ | 5.67 | 0.40 | 0.97↓ | 2.44 |

*Table 6.* Error distribution over Transform candidate samples. Counts are candidate-level counts over the k=5 generation pool for each model; categories are not mutually exclusive, so rows do not sum to the denominator. Structural categories are counted for candidates that fail parsing or LLVM verification. Semantic Mismatch is counted only for verifier-clean candidates that reach equivalence checking and fail semantic validation.

| | GPT | Claude | Gemini | Grok | Qwen | Deepseek |
|---|---|---|---|---|---|---|
| Semantic Mismatch | 59 | 69 | 20 | 50 | 44 | 45 |
| Type and operand | 60 | 88 | 140 | 19 | 72 | 260 |
| Control flow | 428 | 212 | 192 | 40 | 96 | 76 |
| Symbol resolution | 137 | 85 | 120 | 91 | 147 | 129 |
| Number discipline | 34 | 1276 | 52 | 249 | 510 | 700 |

The radar plot in Figure 7 summarizes cross-track behavior. GPT-5 forms the most balanced profile, leading on most axes; Claude/Gemini/Grok cluster as a second tier strong on IR understanding and Repair but clearly weaker on pass-accurate Refactor and Transform; Qwen3-Max-Preview and Deepseek-V3.2-Exp show the sharpest drop when moving from understanding to rewriting. Beyond ranking, three messages stand out: (i) CIRBench cleanly separates IR understanding from IR editing; (ii) semantics-preserving, performance-aware Transform is the bottleneck for all models; and (iii) different families exhibit distinct risk–reward styles, trading fewer miscompilations for fewer performance wins.

## 5. Conclusion

We introduced CIRBench, a layered correctness-aware benchmark for evaluating LLMs on LLVM IR across four compiler-oriented tracks: Analysis, Repair, Refactor, and Transform. CIRBench evaluates whether models can operate at the compiler IR boundary under verifier checks, semantic validation, and performance measurement, rather than merely producing plausible IR text.

Our experiments on six mainstream LLMs show that current models can recover some coarse IR properties and repair a nontrivial fraction of invalid IR, especially when com-piler diagnostics are available. However, reliability drops on tasks requiring precise alias reasoning, control-flow consistency, pass-level invariants, and semantics-preserving optimization. The persistent Valid–Equiv gap shows that verifier-clean IR is still far from semantically reliable IR. On Transform, equivalence-clean candidates occasionally produce large speedups over -O3, but median dynamic performance remains below the compiler baseline.

These results suggest that LLMs are currently better viewed as speculative IR rewrite generators than as standalone optimizing compilers. CIRBench provides a reproducible yardstick for this setting and highlights future directions: graph- and alias-aware representations, pass-level controllability, tighter integration with compiler diagnostics, and risk-aware selection mechanisms for deciding when model-generated IR should be trusted, discarded, or passed back to a classical optimizer.

## Acknowledgement

This work was supported by the Major Science and Technology Project of Jiangsu Province under Grant BG2025011.

## Impact Statement

This work introduces CIRBench, a standardized benchmark for evaluating LLM-guided compiler optimization at the LLVM IR level under layered correctness checks. By surfacing verifier failures, semantic mismatches, and performance regressions, CIRBench can improve the reproducibility and safety of empirical studies in ML for compilers.

At the same time, CIRBench should not be interpreted as a deployment certificate for LLM-generated IR. Passing the LLVM verifier is not semantic safety, and checksum-based fallback testing is not a proof of equivalence. Systems trained or selected using CIRBench could still produce miscompilations when applied outside the benchmark distribution, under different LLVM versions, or on different hardware targets. We therefore recommend that downstream

compiler agents keep verifier, equivalence, testing, and runtime rollback mechanisms in the loop rather than directly trusting model-generated IR.

Better optimization can reduce compute and energy use, but it can also accelerate harmful or abusive workloads. CIRBench is application-agnostic and does not target any specific domain; responsible deployment remains with downstream practitioners and organizations.

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

# Appendix Contents

# A. Related Work

This appendix expands the brief discussion in Section 2 and situates CIRBench within three lines of prior work:

1. IR-level learning and compiler ML.

2. Code and program benchmarks.

3. Correctness-aware evaluation for LLMs

## A.1. IR-level learning and compiler ML

A long line of work applies learning to compiler decision problems without changing the underlying intermediate representation. Classical examples learn heuristics for inlining (Trofin et al., 2021), register allocation (VenkataKeerthy et al., 2023), or pass ordering on top of existing LLVM or GCC pipelines (Pan et al., 2025b; Zhu et al., 2024; Gao et al., 2025), typically treating the compiler as a black-box environment and keeping optimizations themselves hand-written (Seeker et al., 2024). Correctness in these systems is inherited from the trusted passes: learned policies decide *when* to run a pass or which parameter setting to choose, but do not construct new IR (Singh et al., 2024; Zhu & Hao, 2023). CIRBench is complementary: it assumes the compiler infrastructure as given and instead focuses on how well large models can operate *inside* the middle end, at the IR boundary where passes usually act.

Another strand studies IR representations for downstream prediction or cost modelling. Graph- and sequence-based encoders over LLVM IR, such as ProGraML (Cummins et al., 2021), MIR-based encoders (Dutta & Jannesari, 2024), and IR2Vec (VenkataKeerthy et al., 2020), have been used to predict performance, energy, and optimization profitability, or to drive autotuning and hardware mapping (Rotem & Cummins, 2021). These works treat IR mainly as an input signal and evaluate models with accuracy or regression error on scalar labels (e.g., speedup class, vectorizability (Wu et al., 2022; Taneja et al., 2025), or phase choice). CIRBench reuses similar IR front-ends but targets a different regime: multi-step reasoning and *editing* of IR, where the outputs are new IR programs that must satisfy compiler-level semantic constraints.

Closer to our setting, neural and symbolic superoptimization approaches directly propose IR rewrites. Souper (Sasnauskas et al., 2017) and related SMT-based tools (Schkufza et al., 2013; 2014) search locally for better LLVM IR fragments under formal equivalence checking (Sharma et al., 2015), while equality-saturation systems explore large equivalence classes using e-graphs and extract cost-optimal representatives (Willsey et al., 2021; Fan & Regehr, 2024). Recent work uses neural models, including LLMs, to generate candidate peephole rules or to steer the exploration (Taneja et al., 2025; Pan et al., 2025a). These systems typically evaluate on internally curated sets of rewrite tasks, focusing on rule discovery or local speedups. CIRBench is not a superoptimizer: instead, it exposes a standardized collection of IR analysis, repair, refactor, and transform tasks, designed to be reusable across model architectures and to decouple model evaluation from any specific optimization engine.

Finally, several recent papers prompt LLMs directly on LLVM IR or similar low-level forms for code understanding, bug repair, or optimization (Gong et al., 2025; Jiang et al., 2025; Italiano & Cummins, 2025; Zhang et al., 2025a; Yang et al., 2026). Each of these studies proposes its own tasks (e.g., explaining IR in natural language, predicting vectorizability, or rewriting to approach `-O3`) and metrics, often mixing source-level and IR-level supervision. By contrast, CIRBench aims to provide a shared, IR-native benchmark with layered correctness-aware metrics, so that results across such systems become comparable and can accumulate over time.

## A.2. Code and program benchmarks

The rapid progress of code-oriented LLMs has been driven in part by a rich ecosystem of benchmarks at the source-code level. HumanEval (Chen, 2021), MBPP (Austin et al., 2021), CodeXGLUE (Lu et al., 2021), APPS (Hendrycks et al., 2021), and related datasets evaluate functional correctness of short programs from natural-language specifications using unit tests, typically in Python, C++, or Java. Subsequent benchmarks broaden the task space to include code translation, type inference, refactoring, and program repair, often across multiple languages and with larger, more realistic problems (Kulal et al., 2019; Coignion et al., 2024; Li et al., 2023; Roziere et al., 2023). More recently, system-level benchmarks such as SWE-bench and long-horizon repository tasks probe end-to-end software engineering workflows, where models interact with issue trackers, test suites, and version control (Jimenez et al., 2023; Wu et al., 2024).

These benchmarks share two properties. First, they operate on high-level source code, where syntax and naming conventions

carry human intent and where tests are readily available. Second, they typically evaluate only functional correctness (does the program pass the tests?) and sometimes style or documentation quality, but rarely expose compiler-internal notions such as UB, IR attributes, or fine-grained performance effects of low-level rewrites. As a result, code benchmarks are well suited for measuring general programming ability, but less so for understanding how models behave as components *inside* optimizing compilers.

CIRBench is complementary to source-level benchmarks. By fixing the representation to LLVM IR and using verifier/equivalence checks plus performance measurements, it abstracts away lexical choices and focuses on the semantics and cost of IR-level edits. We view CIRBench as a middle layer between source-level code benchmarks and microarchitectural simulators: it retains enough structure to be compiler-relevant, while remaining independent of any particular front-end language or hardware design.

### A.3. Correctness-aware evaluation for LLMs

Beyond compilers, correctness has become a central concern in evaluating LLMs for code and formal reasoning. Many code benchmarks adopt test-based metrics such as Pass@k, which account for sampling variation and encourage diverse candidate generation (Lu et al., 2021; Austin et al., 2021). Subsequent work augments test suites with mutation-based or fuzzing-based oracles, or integrates static analyzers to detect undefined behavior and security issues beyond what unit tests cover (Roziere et al., 2021; Sandoval et al., 2023; Yao et al., 2024). In parallel, LLM applications to theorem proving, program verification, and program synthesis have explored combining models with SMT solvers, proof assistants, and abstract interpreters to enforce or check logical correctness (Bansal et al., 2019; Song et al., 2024; Gupte et al., 2025).

In the compiler community, formal correctness has long been studied through verified compilers and translation validation. Projects such as CompCert and CakeML prove whole pipelines correct with respect to formal semantics of source and target languages (Leroy, 2009; Kumar et al., 2014), while tools like Alive and Alive2 reason about the correctness of individual LLVM optimizations, either off-line or at JIT time (Lopes et al., 2015; 2021). Recent LLM-for-compilers work sometimes borrows these tools to validate model-proposed optimizations, but usually in bespoke setups without a common benchmark.

CIRBench can be seen as bringing these strands together. It adopts a *layered* correctness-aware protocol—combining the LLVM verifier, function-level equivalence checking, and checksum-based execution tests—and exposes this protocol as part of a reusable benchmark rather than as an internal implementation detail. By reporting Valid@k and Equiv@k alongside performance, and by categorizing failure modes, CIRBench encourages evaluation practices that go beyond aggregate speedups or bare EM scores. We hope this will make it easier to compare LLM-based compiler components, to stress-test future models under realistic correctness constraints, and to transfer ideas from formal verification and testing into IR-level LLM evaluation.

## B. Prompt Templates and I/O Formats

We use deliberately simple, uniform prompts across all tracks to establish a conservative baseline. Prompts never leak ground-truth answers, pass sequences, or oracle outputs; they only describe the task, the input format, and the expected output schema. We also avoid chain-of-thought instructions or tool-calling hints. This design is intentional: we want CIRBench to serve both as a benchmarking suite *and* as a testbed for prompt engineering and specialized agents, on top of a clean reference setting.

Across tracks we adopt a few global conventions:

- IR payloads are wrapped in explicit tags such as `<IR>...</IR>` and must be echoed back unchanged.

- Generated IR must be wrapped in `<IR_OUT>...</IR_OUT>`; analysis answers use `<CIR_JSON>...</CIR_JSON>`.

- Any text outside the requested wrappers is ignored.

Below we outline the textual structure of the prompts and give one full example per track. Complete prompt files, including all modes and minor variants, are part of the released toolkit.

## B.1. Analysis Track Prompts

Analysis prompts frame the model as a static-analysis assistant. Each instance contains (i) a verbatim LLVM IR function wrapped in <IR>...</IR> and (ii) a natural-language question describing which analysis to perform (CFG, Alias, DomTree, Loops, or Properties). The prompt then specifies the JSON schema and requests that the answer be wrapped between <CIR_JSON> and </CIR_JSON>.

Below we show the full prompt and answer format for the Alias subtask; other subtasks follow the same pattern with different JSON fields.

**Alias example.**

```
You are an LLVM IR analysis assistant.
Given LLVM IR:
<IR>
define void @foo() {
  %G26 = getelementptr i1, ptr undef, i1 undef
  %B20 = shl i8 -128, 16
  %G47 = getelementptr ptr, ptr undef, i8 %B20
  load i1, ptr %G26
  load ptr, ptr %G47
  ret void
}
</IR>

Analyze the alias info of the IR.
Output ONLY JSON as:
{"labels":[{"pair":["%G26","%G47"],"value":"?"}]}

Rules for alias analysis:
- "No": The two pointers never alias (point to disjoint memory locations)
- "Must": The two pointers always alias (point to the same memory location)
- "May": The two pointers may alias in some executions (cannot be determined statically)
- "Partial": The pointers may partially overlap

Wrap it between <CIR_JSON> ... </CIR_JSON>.
```

A valid model answer is:

```
<CIR_JSON>
{
  "labels": [
    {
      "pair": ["%G26", "%G47"],
      "value": "NoAlias"
    }
  ]
}
</CIR_JSON>
```

Prompts for CFG, DomTree, Loops, and Properties mirror this structure, differing only in the question sentence and the JSON fields ("edges", "dom", "loops", "stats", etc.). During evaluation we strip the <CIR_JSON> wrapper, parse the payload as JSON, and apply canonicalization (key ordering, set sorting, numeric normalization) before computing EM and F1. For F1, we treat each JSON list field as an unordered set of atomic items (e.g., alias labels, CFG edges), and compute precision/recall by set overlap between the predicted and reference sets. **We then report the micro-averaged F1 over all instances.**

## B.2. Repair Track Prompts

Repair prompts describe the task as "fix the IR if it is wrong" and ask the model to output an entire define block wrapped in <IR_OUT>...</IR_OUT>. Both modes explicitly forbid changing function signatures, linkage, attributes, or adding extra helper functions.

In the *Hint* mode, the verifier error message and source location are appended after the IR; in the *Hard* mode, only the IR and a generic instruction are given. Below is a representative Hint-mode prompt:

```
You are an LLVM IR repair assistant.
Fix the IR if it fails assembly or verifier; otherwise reply exactly: No
Hint:
- error: '%11' defined with type 'i32' but expected 'i16' : %26 = trunc i16 %11 to i8
Begin IR:
<IR>
define dso_local void @foo(ptr noundef %p) {
  ; IR body with errors
}
</IR>

Verifier error:
line 4: PHI node has incoming value with wrong type

Output format:
- If a repair is needed, output ONLY the repaired function as:
  <IR_OUT>
  define ...
    ...
  }
  </IR_OUT>
- Otherwise output the single word: No
```

Hard-mode prompts omit the `Verifier error` block and keep the rest identical.

## B.3. Refactor Track Prompts

Refactor-Normal prompts identify a single optimization pass in natural language and request that the model apply *only* this pass, again returning a full IR function wrapped in `<IR_OUT>...</IR_OUT>`.

**Refactor-Normal example.**
```
You are an LLVM IR refactor assistant.
Apply exactly this optimization pass, no others: CorrelatedValuePropagation
IR:
<IR>
; unoptimized LLVM IR for function @foo
define i32 @foo(i32 %x, i32 %y) {
entry:
  %cmp = icmp eq i32 %x, %y
  br i1 %cmp, label %then, label %else
then:
  %a = add i32 %x, 1
  br label %merge
else:
  %b = add i32 %x, 1
  br label %merge
merge:
  %phi = phi i32 [ %a, %then ], [ %b, %else ]
  ret i32 %phi
}
</IR>

If the given pass would make no change, reply exactly: No.
Otherwise, output ONLY the transformed function as:
<IR_OUT>...define @foo(...) { ... }</IR_OUT>
```

Refactor-Reverse prompts instead present a before/after IR pair and ask the model to output only the pass name from a closed list:

```
I have applied ONE LLVM pass to transform BEFORE LLVM IR into AFTER LLVM IR.
Identify which pass was applied from the following list, and output ONLY JSON.
```

```
List of candidate passes:
- CorrelatedValuePropagation
- DeadStoreElim
- EarlyCSE
- GlobalOpt
- GVN
- IndVarSimplify
- InstCombine
- InstSimplify
- IPSCCP
- JumpThreading
- LCSSA
- LICM
- LoopRotate
- LoopSimplify
- LoopUnroll
- MemCpyOpt
- Reassociate
- SimplifyCFG
- SROA
- TailCallElim

Return ONLY the pass name in this format without any explanation: <CIR_JSON>{"pass":"<ONE_OF_LIST>"}</CIR_JSON>
BEFORE:
<BEFORE_IR>
  ; original IR for @foo
</BEFORE_IR>

AFTER:
<AFTER_IR>
  ; IR after applying one pass
</AFTER_IR>
```

## B.4. Transform Track Prompts

Transform prompts describe the task as performance-oriented but semantics-preserving IR optimization. Each prompt provides a -O0 IR fragment, a short note about the target architecture, and the requirement that the result must be functionally equivalent. In *Direct* mode, the model returns a single optimized IR version; in *Copilot* mode, the prompt text is identical but the evaluation pipeline subsequently runs -O3 on the model output. The MODE is dynamic selected by the toolchain, and the corrspondence prompt is applied.

```
You are an LLVM IR transform assistant.

MODE = {MODE}

If MODE == "function":
- Target function: {FUNC_NAME}
- Apply O3-style IR-level optimizations that LLVM would reasonably perform.
- Preserve module-level globals, declarations, and attributes as-is.
- Return ONLY the transformed function (from 'define' to the matching '}') wrapped in <IR_OUT>...</IR_OUT>.
- If no change is needed, reply exactly: No

If MODE == "module":
- Apply O3-style IR-level optimizations to the entire module.
- Preserve declarations and non-semantic metadata that do not affect code generation.
- Return ONLY the transformed module wrapped in <IR_OUT>...</IR_OUT>.
- If no change is needed, reply exactly: No

Return format:
- EXACTLY ONE block: <IR_OUT>...LLVM IR...</IR_OUT> (or the single word: No)

Given IR:
```

```
<IR>
  ; unoptimized -O0 IR for @kernel_run
</IR>
```

## C. Analysis Track Details

### C.1. Subtasks and labels

The ANALYSIS track in CIRBench is designed to probe whether a model can recover core compiler-side facts from LLVM IR, rather than only generate code. We decompose the track into five primary subtasks, A1–A5 (internally A001–A005), each corresponding to a family of standard LLVM analysis. For every subtask, we derive ground truth from the reference compiler and expose it as a labeling or prediction problem over IR.

**A1 / A001_Alias: alias.** This subtask targets pointer aliasing information. It is grounded in LLVM's alias analysis framework (`BasicAA`, `CFLAA`, and related passes). Given pairs of memory-accessing instructions (loads, stores, calls) and pointer expressions extracted from a function, the model must classify their relationship as *NoAlias*, *MayAlias*, *PartialAlias*, or *MustAlias*. We obtain labels by running LLVM's alias queries on the unmodified IR and normalizing the results to the above finite vocabulary. Alias information is a prerequisite for many downstream transformations (e.g., loop-invariant code motion, store-to-load forwarding, dead store elimination), so this subtask provides a direct test of IR-level memory reasoning.

**A2 / A002_Loops: loop structure.** A2 focuses on loop detection and nesting, in line with LLVM's loop analysis passes (e.g., `-loop-info`, `-loop-simplify`, `-scalar-evolution`). For each function, we ask the model to identify loop headers and latches, group basic blocks into natural loops, and recover simple loop nesting structure. Ground truth is derived from `LoopInfo` and related analyses. This subtask probes whether models can reconstruct loop structure and simple recurrence patterns from raw IR, which is essential for unrolling, vectorization, and software pipelining.

**A3 / A003_DomTree: dominance.** A3 addresses dominance relations and dominator-tree topology, reflecting the behavior of LLVM's `-domtree` analysis. We represent the dominator tree as a json list of records of the form {`"block": <name>`, `"depth": <int>`, `"idom": <name or null>`}. Given this representation, the model must reconstruct the full tree: for each basic block, it predicts its depth in the tree and its immediate dominator (or `null` for the entry block). The subtask thus measures how well a model internalizes the control dependencies encoded by IR.

**A4 / A004_CFG: control-flow skeleton.** A4 focuses on general control-flow graph (CFG) properties beyond pure dominance. It is aligned with passes such as CFG simplification (`-simplifycfg`), unreachable code elimination, and various reachability analyses. We encode the CFG as json containing lists of predecessors for each basic block, together with reachability information from the entry block. Tasks include reconstructing the successor/predecessor lists and identifying unreachable blocks. Ground truth is extracted from the compiler's CFG utilities. This subtask stresses whether models can reconstruct a sound control-flow skeleton and reason about which paths are executable.

**A5 / A005_Properties: quantitative IR properties.** A5 collects simple quantitative and structural properties of IR functions. For each instance we ask for counts such as the number of basic blocks, instructions, loads, stores, and calls. We encode answers as a flat json object mapping property names (e.g., `"num_blocks"`, `"num_loads"`) to integer values. Ground truth is computed by a deterministic IR walker that mirrors how compilers maintain summary statistics about functions. This subtask is intentionally easier than A1–A4, but complements them by checking whether models keep track of simple global invariants that later matter for Repair and Transform.

### C.2. Prevalence statistics

The "Prevalence" column in Table 2 reports how often each analysis category is used inside LLVM's transformation pipeline. To estimate these frequencies, we perform a static pass-level analysis over LLVM's transformation library:

- We scan the source code of transformation passes under `llvm/lib/Transforms` for explicit analysis dependencies, such as `getAnalysis<AAResultsWrapperPass>()`, `LoopInfo`, `DominatorTree`, and CFG utilities.

- For each pass, we annotate a binary vector indicating whether it depends on alias information (A1), loop information (A2), dominator tree information (A3), or CFG structure (A4).

- We then aggregate these annotations over all passes in the default `-O3` pipeline and report, for each category, the fraction of passes that depend on at least one analysis in that family.

These percentages are not meant as exact measures of dynamic importance, but rather as a coarse indication that A1–A4 correspond to analyses that are widely reused across optimization passes. A5 (PROPERTIES) measures generic statistics and is therefore not tied to a particular analysis pass, so we omit a prevalence score for it.

### C.3. Difficulty and IR scale

As described in subsection 3.1, we stratify Analysis instances into Easy, Medium, and Hard levels based on IR size and structural complexity. Here we illustrate how this plays out in practice using the dominator-tree representation from A3.

For a simple, shallow function, the oracle dominator tree may look like:

```
{
  "labels": [
    { "block": "%b0", "depth": 1, "idom": null },
    { "block": "%b3", "depth": 2, "idom": "%b0" },
    { "block": "%b4", "depth": 2, "idom": "%b0" }
  ]
}
```

In contrast, Hard instances can involve dozens of basic blocks and deep nesting, such as:

```
{
  "labels": [
    { "block": "%1",  "depth": 1,  "idom": null },
    { "block": "%15", "depth": 2,  "idom": "%1" },
    { "block": "%16", "depth": 3,  "idom": "%15" },
    { "block": "%34", "depth": 4,  "idom": "%16" },
    { "block": "%.preheader36", "depth": 5, "idom": "%34" },
    { "block": "%45", "depth": 6,  "idom": "%.preheader36" },
    { "block": "%47", "depth": 7,  "idom": "%45" },
    { "block": "%48", "depth": 8,  "idom": "%47" },
    { "block": "%50", "depth": 9,  "idom": "%48" },
    { "block": "%54", "depth": 10, "idom": "%50" },
    { "block": "%56", "depth": 11, "idom": "%54" },
    { "block": "%38", "depth": 12, "idom": "%56" },
    { "block": "%64", "depth": 13, "idom": "%38" },
    { "block": "%69", "depth": 14, "idom": "%64" },
    { "block": "%.preheader", "depth": 15, "idom": "%69" },
    { "block": "%74", "depth": 16, "idom": "%.preheader" },
    { "block": "%76", "depth": 17, "idom": "%74" },
    { "block": "%77", "depth": 18, "idom": "%76" },
    { "block": "%79", "depth": 19, "idom": "%77" },
    { "block": "%85", "depth": 20, "idom": "%79" },
    { "block": "%87", "depth": 21, "idom": "%85" },
    { "block": "%71", "depth": 22, "idom": "%87" },
    { "block": "%92", "depth": 20, "idom": "%79" },
    { "block": "%94", "depth": 19, "idom": "%77" },
    { "block": "%90", "depth": 19, "idom": "%77" },
    { "block": "%88", "depth": 17, "idom": "%74" },
    { "block": "%72", "depth": 14, "idom": "%64" },
    { "block": "%61", "depth": 10, "idom": "%50" },
    { "block": "%63", "depth": 9,  "idom": "%48" },
    { "block": "%59", "depth": 9,  "idom": "%48" },
    { "block": "%57", "depth": 7,  "idom": "%45" },
    { "block": "%95", "depth": 4,  "idom": "%16" },
    { "block": "%43", "depth": 4,  "idom": "%16" },
    { "block": "%96", "depth": 3,  "idom": "%15" },
    { "block": "%32", "depth": 3,  "idom": "%15" },
    { "block": "%97", "depth": 2,  "idom": "%1" },
    { "block": "%30", "depth": 2,  "idom": "%1" }
  ]
}
```

The latter requires the model to keep track of over thirty basic blocks, multiple nested loops, and non-trivial join points. Similar scaling trends apply to the other subtasks: for CFG, the number of edges and branching factors increase; for aliasing, the number of pointer pairs and memory-access sites grows; and for A5, the counts span larger ranges. Difficulty labels (Easy/Medium/Hard) are assigned by thresholds on instruction count, basic-block count, loop nesting depth, and analysis-specific structural signals, chosen to roughly align with terciles of the underlying distributions (Appendix G).

## D. Refactor Track Details

### D.1. Refactor passes (RF001–RF020)

The REFACTOR track is meant to test whether a model can reproduce the kind of small, semantics-preserving cleanups that real compilers apply all the time: canonicalizing loops, simplifying expressions, removing dead work, and reshaping control flow without changing observable behavior. Concretely, we instantiate micro-tasks around 20 classic LLVM passes. Table 7 lists the passes and their corresponding new-pass-manager names; below we briefly summarize what each pass does. For a few representative passes we also show small IR snippets to make the effect more concrete.

*Table 7.* Refactor passes used in CIRBench. Each RF code corresponds to a single LLVM pass in the new pass manager.

| ID | LLVM pass (informal name) | NewPM pass name (for `-passes`) |
|----|----------------------------|----------------------------------|
| RF001 | Correlated Value Propagation | `correlated-propagation` |
| RF002 | Dead Store Elimination | `dse` |
| RF003 | Early CSE | `early-cse` |
| RF004 | Global Optimizations | `globalopt` |
| RF005 | Global Value Numbering | `gvn` |
| RF006 | Induction Variable Simplify | `indvars` |
| RF007 | Instruction Combine | `instcombine` |
| RF008 | Instruction Simplify | `instsimplify` |
| RF009 | Interprocedural SCCP | `ipsccp` |
| RF010 | Jump Threading | `jump-threading` |
| RF011 | Loop-Closed SSA | `lcssa` |
| RF012 | Loop-Invariant Code Motion | `licm` |
| RF013 | Loop Rotate | `loop-rotate` |
| RF014 | Loop Simplify | `loop-simplify` |
| RF015 | Loop Unroll | `loop-unroll` |
| RF016 | MemCpy Optimization | `memcpyopt` |
| RF017 | Reassociate | `reassociate` |
| RF018 | Simplify CFG | `simplifycfg` |
| RF019 | Scalar Replacement of Aggregates | `sroa` |
| RF020 | Tail Call Elimination | `tailcallelim` |

**RF001_CorrelatedValuePropagation.** `CorrelatedValuePropagation` propagates information learned from branches or comparisons into later uses of the same value. If a branch guards a block with a known condition (e.g., if (x == 0) ... ), subsequent checks that are always true or false on that path can be simplified away and now-unreachable code can be removed.

```
; Before CVP: the second compare is redundant once we know %x == 0.
define void @rf001(i32 %x) {
entry:
  %cmp  = icmp eq i32 %x, 0
  br i1 %cmp, label %then, label %else

then:
  %cmp2 = icmp eq i32 %x, 0   ; always true on this path
  br i1 %cmp2, label %hot, label %cold
hot:
  ret void
cold:
  ret void
else:
  ret void
}
```

**RF002_DeadStoreElim.** `DeadStoreElim` looks for stores whose values are never read before being overwritten or discarded (e.g., stored again or the stack frame is deallocated) and removes such dead stores, reducing memory traffic without changing semantics.

```
; The first store to %p can be removed: it is overwritten before any load.
```

```llvm
define void @rf002(i32* %p) {
entry:
  store i32 0,  i32* %p
  store i32 42, i32* %p
  ret void
}
```

**RF003 EarlyCSE.**    `EarlyCSE` (early common subexpression elimination) removes redundant computations within a basic block or simple region. It detects when instructions compute the same value from the same operands (including some loads) and replaces later ones with earlier ones. This pass runs early, before more aggressive global optimizations, and also eliminates trivial redundant loads when memory is provably unchanged.

**RF004 GlobalOpt.**    `GlobalOpt` performs simple whole-program global-variable optimizations. Typical transformations include converting some globals into constants, promoting private globals used in only one function into locals or direct scalars, and removing unused initializers. These changes simplify the IR and often expose further constant folding and dead-code elimination.

**RF005 GVN.**    `GVN` (global value numbering) is a more global form of common subexpression elimination. It assigns "value numbers" to computations, identifies equivalent expressions across basic blocks, replaces later redundant computations with earlier ones, and can eliminate loads dominated by stores to the same address when there are no intervening clobbers.

**RF006 IndVarSimplify.**    `IndVarSimplify` canonicalizes loop induction variables. It rewrites loops so that they use a single, well-behaved induction variable (typically counting up from a start to an end) and normalizes comparisons and step sizes. This makes subsequent loop passes (unrolling, vectorization, etc.) easier to apply.

**RF007 InstCombine.**    `InstCombine` is LLVM's main peephole and algebraic-simplification pass. It looks at small instruction patterns and replaces them with simpler, equivalent forms: add x, $0 \rightarrow$ x, mul x, $1 \rightarrow$ x, merging chains of casts, folding arithmetic with constants, and combining compares.

```llvm
; Before InstCombine: trivial arithmetic and cast chains.
define i32 @rf007(i32 %x) {
entry:
  %a = add i32 %x, 0        ; -> %x
  %b = mul i32 %a, 1        ; -> %a
  %c = zext i32 %b to i64
  %d = trunc i64 %c to i32    ; zext+trunc round-trips to %b
  ret i32 %d               ; ideally just "ret i32 %x"
}
```

**RF008 InstSimplify.**    `InstSimplify` is a lightweight library of simplification rules used by many passes. Given constant or known-equal operands it simplifies instructions without constructing new ones, e.g., icmp ne x, x $\rightarrow$ false . In CIRBench, we expose it as an explicit task to test whether models can emulate these logical simplifications.

**RF009 IPSCCP.**    `IPSCCP` (interprocedural sparse conditional constant propagation) propagates constant values across function boundaries. It tracks which arguments and return values are constant under the current call graph, replaces uses with constants, and deletes branches that become unreachable, achieving many of the benefits of "specialization" without full code cloning.

**RF010 JumpThreading.**    `JumpThreading` rewrites control flow when branch outcomes can be predicted along certain paths. If a block ends with a branch whose outcome is known for some predecessors, the pass can "thread" those predecessors directly to the known successor, collapsing diamonds and reducing branch overhead.

**RF011 LCSSA.**    `LCSSA` (loop-closed SSA) ensures that any value defined inside a loop and used outside passes through a PHI node on an exit block. This form makes it easier for loop passes to update SSA when duplicating or peeling loops, because all cross-loop uses are explicit at the exits.

**RF012_LICM.**    `LICM` (loop-invariant code motion) hoists loop-invariant computations out of loops and sinks code that only affects dead paths into loop exits. If an instruction produces the same result on every iteration and is safe to execute once, it can be moved to the preheader.

```
; The address computation %p is loop-invariant and can be hoisted.
define void @rf012(i32* %base, i32 %n) {
entry:
  br label %loop

loop:
  %i    = phi i32 [ 0, %entry ], [ %next, %loop ]
  %p    = getelementptr inbounds i32, i32* %base, i32 %i
  store i32 %i, i32* %p
  %next = add i32 %i, 1
  %cond = icmp slt i32 %next, %n
  br i1 %cond, label %loop, label %exit

exit:
  ret void
}
```

**RF013_LoopRotate.**    `LoopRotate` changes loop shape to a more canonical "rotated" form, typically turning a loop with an early exit test into one whose header always executes and has a single backedge. This exposes preheaders, improves fall-through layout, and helps passes such as unrolling and vectorization that expect canonicalized control flow.

**RF014_LoopSimplify.**    `LoopSimplify` enforces the "simplified loop form" used throughout LLVM: each loop has a unique preheader, a single latch (backedge), and well-formed exit blocks. It may split blocks to isolate headers and exits, making loop structure explicit.

**RF015_LoopUnroll.**    `LoopUnroll` replicates the loop body multiple times per iteration, reducing branch overhead and exposing more straight-line code. It can perform full unrolling or partial/unroll-and-jam depending on heuristics and trip-count information.

```
; Conceptually, this loop:
;   for (i = 0; i < 4; ++i) body(i);
define void @rf015(i32* %p) {
entry:
  br label %loop
loop:
  %i    = phi i32 [ 0, %entry ], [ %next, %loop ]
  %addr = getelementptr inbounds i32, i32* %p, i32 %i
  store i32 %i, i32* %addr
  %next = add i32 %i, 1
  %cond = icmp slt i32 %next, 4
  br i1 %cond, label %loop, label %exit
exit:
  ret void
}
```

After unrolling by 2, the body would be duplicated and the induction variable updated accordingly, reducing the number of loop trips.

**RF016_MemCpyOpt.**    `MemCpyOpt` optimizes memory copy/move/set operations. It can replace small `memcpy/memmove` calls with explicit loads and stores, merge adjacent stores into wider stores, or remove redundant copies when source and destination are already equal.

**RF017_Reassociate.**    `Reassociate` rewrites chains of associative/commutative operations (such as addition and multiplication) into a different tree shape that is more amenable to constant folding or common-subexpression elimination. For instance, it may transform (a + b) + (c + 1) into a + b + c + 1, then fold constants and share repeated subexpressions.

**RF018_SimplifyCFG.** `SimplifyCFG` simplifies the control-flow graph. It merges blocks with single predecessors/successors, folds branches with constant conditions, simplifies `switch` statements, and removes unreachable blocks, producing a more compact CFG.

**RF019_SROA.** `SROA` (scalar replacement of aggregates) breaks down aggregate objects (structs, arrays) stored in memory into independent scalar values when possible. Applied to `allocas` and certain globals, it turns loads/stores of subfields into direct scalar SSA values, generalizing `mem2reg`.

```
; Before SROA: struct in memory.
%Pair = type { i32, i32 }

define i32 @rf019() {
entry:
  %p    = alloca %Pair
  %xptr = getelementptr inbounds %Pair, %Pair* %p, i32 0, i32 0
  %yptr = getelementptr inbounds %Pair, %Pair* %p, i32 0, i32 1
  store i32 1, i32* %xptr
  store i32 2, i32* %yptr
  %x    = load i32, i32* %xptr
  %y    = load i32, i32* %yptr
  %s    = add i32 %x, %y
  ret i32 %s
}
```

After SROA, the struct never exists in memory: %x and %y become pure SSA scalars.

**RF020_TailCallElim.** `TailCallElim` transforms tail-recursive calls into equivalent loops or marks eligible calls as tail calls. In the recursive case, a function that calls itself in tail position can be rewritten into a loop that reuses the same stack frame, avoiding unbounded recursion and reducing call overhead.

Taken together, RF001–RF020 cover a broad slice of "everyday" refactoring passes in LLVM: from local algebraic simplifications, through SSA and CFG canonicalization, to loop and memory cleanups. The REFACTOR track uses focused tasks around these passes to evaluate whether models understand not just what IR *means*, but also how a production compiler expects it to be shaped.

### D.2. Instance construction

We construct Refactor instances automatically from existing LLVM IR using the following pipeline.

**Collecting before/after pairs.** Starting from IR-Optset and additional IR pools described in Appendix H, we run

```
opt -print-changed -print-before-changed -passes="default<O3>" --print-module-scope input.ll
```

and parse the log. For each pass in RF001–RF020, we extract all segments where that pass reports a modification, yielding a set of *before/after* IR pairs for that pass.

**Filtering and equivalence checking.** Many such pairs are either too trivial (e.g., only debug metadata changes) or too large for our token budgets. We therefore filter candidates using three criteria: (i) the textual IR must change by at least a small threshold (number of modified instructions); (ii) the function size after normalization must fall within the Easy/Medium/Hard ranges in Appendix G; and (iii) the before/after pair must be semantically equivalent according to our equivalence oracle. Equivalence is checked using Alive2 at function level; pairs that fail verification or trigger tool timeouts are discarded.

**Sampling final instances.** For each of the 20 passes we randomly sample 10 accepted pairs, stratified by Easy/Medium/Hard to obtain a balanced difficulty distribution ('10 per pass, 200 total' in the main text). For Refactor-Normal, the *before* IR serves as the prompt input, the pass name is given, and the *after* IR is the reference target. For Refactor-Reverse, we present both the before and after IR and ask the model to identify which of the 20 passes produced the transformation.

Because every instance is derived from a single-pass change in the `-print-changed` log and validated by equivalence checking, we avoid ambiguity with multi-pass or composite transformations.

### D.3. Normalization

To focus the Refactor tasks on *semantic* and *structural* changes rather than on arbitrary naming or formatting differences, we apply a deterministic normalization procedure to both the before and after IR, following the conventions of the IR-Optset toolchain.

**Identifier and block renaming.** We rename functions, basic blocks, and SSA values to a canonical scheme (e.g., @STRUCT1, BB0, %0, %1, …) based on first occurrence. This removes accidental cues from human-chosen symbol names and makes EM comparison less sensitive to irrelevant renaming.

**Metadata and attribute handling.** Non-semantic metadata (debug info, `!dbg` tags) is stripped. Semantic attributes (e.g., `nuw`, `readonly`, calling convention flags) are preserved and printed in a canonical order. Landing pads, personality functions, and other exception-related constructs are retained only when semantically required.

**Formatting and ordering.** We normalize indentation, remove redundant whitespace, and emit instructions in a fixed order within each basic block. Where LLVM allows reordering of commutative operands or PHI incoming edges, we sort them by a stable key (e.g., SSA name) so that equivalent IR prints identically. This normalization is applied symmetrically to model outputs before computing EM, and to reference IR when we generate the dataset.

The combined construction and normalization pipeline ensures that Refactor instances are (i) genuine single-pass transformations with verified semantic equivalence, and (ii) presented in a way that rewards models for matching the intended pass behavior, not for guessing superficial naming or formatting patterns.

## E. Repair Track Details

### E.1. Repair Error Taxonomy

When large language models generate LLVM IR directly, they tend to miss subtle structural and typing invariants that human-authored IR almost never violate. In CIRBench, the REPAIR track focuses on a set of verifier and parser error signatures that are especially characteristic of LLM-generated IR. For each signature below we give a brief description and a minimal IR fragment that triggers the corresponding error. Actual diagnostics may differ slightly across LLVM versions, and a single fragment can trigger multiple messages; we group examples by the *primary* pattern we target.

**(R1) Inconsistent inferred type for a forward-referenced value.** **Typical message:** `error: '%10' defined with type <type_a> but expected <type_b>`.

The assembler infers the type of a forward-referenced SSA value from its first use. If the later definition uses a different type, the two types disagree and the verifier reports an inconsistency.

```
; %10 is first used as an i32, but later defined as an i64.
define void @r1() {
entry:
  %0  = add i32 %10, 1      ; first use: %10 is assumed to be i32
  %10 = add i64 0, 1        ; definition: %10 produces an i64
  ret void
}
```

**(R2) Forward-referenced instruction with pointer type.** **Typical message:** `error: instruction forward referenced with type 'ptr'`.

An unnamed value is used as a pointer operand before it is defined, so the assembler assumes a pointer type for the forward reference. Later defining it with a non-pointer type, or not defining it at all, yields a forward-reference error.

```
define i32 @r2() {
```

```
entry:
  %0 = load i32, ptr %2        ; %2 is assumed to be a pointer
  %2 = add i32 1, 2            ; later definition is an i32, not a pointer
  ret i32 %0
}
```

**(R3) Non-monotonic numbering of unnamed temporaries.**   **Typical message:** `error: instruction expected to be` `numbered <%X> or greater`.

Within a function, unnamed temporaries (%0, %1, %2, ...) must be assigned monotonically increasing numbers. Using a lower number after a higher one violates this invariant.

```
define i32 @r3(i32 %a) {
entry:
  %0 = add i32 %a, 1
  %2 = add i32 %0, 1           ; first unnamed use after %0 is %2
  %1 = add i32 %0, 2           ; later unnamed value %1 breaks ordering
  ret i32 %1
}
```

**(R4) Integer constant used with a floating-point type.**   **Typical message:** `error: integer constant must have integer` `type`.

This pattern uses an integer literal where a floating-point constant is expected.

```
define double @r4() {
entry:
  %x = fadd double 1.0, 2.0   ; well-typed floating-point add
  %z = fdiv double %x, 1      ; '1' is parsed as an integer, not a double
  ret double %z
}
```

**(R5) Floating-point constant invalid for an integer type.**   **Typical message:** `error: floating point constant invalid` `for type`.

Conversely, a floating-point literal may be used where an integer constant is expected.

```
define i32 @r5() {
entry:
  ; 1.0 is not a valid constant for integer type i32.
  %x = add i32 1.0, 2
  ret i32 %x
}
```

**(R6) Return value type does not match function result type.**   **Typical message:** `error: value doesn't match function` `result type <type>`.

The type of the operand passed to `ret` must match the function's declared result type exactly.

```
; Declared to return i32, but returns a float value instead.
define i32 @r6() {
entry:
  %x = fadd float 1.0, 2.0
  ret float %x                ; type of %x does not match result type i32
}
```

**(R7) Invalid cast opcode for a source/target type pair.**   **Typical message:** `error: invalid cast opcode for cast from` `<type_A> to <type_B>`.

Not all combinations of source/target types are legal for a given cast opcode. For example, zext may only convert from a smaller integer type to a larger integer type.

```
define double @r7() {
entry:
  %x = fadd float 1.0, 2.0
  ; zext is only valid from integer to a wider integer type, not from float.
  %y = zext float %x to double
  ret double %y
}
```

**(R8) Invalid TBAA access type node.**   **Typical message:** Access type node must be a valid scalar type.

TBAA metadata must refer to a valid scalar type node as the access type. Inventing an arbitrary node that does not encode a scalar type leads to this error.

```
define void @r8(i32* %p) {
entry:
  ; The !tbaa attachment refers to an invalid access type node !1.
  %x = load i32, i32* %p, !tbaa !1
  ret void
}

!0 = !{ !"tbaa_root" }
!1 = !{ !"tbaa_root", !"not_a_type" } ; invalid access type node
```

**(R9) TBAA tag attached to an unsupported instruction.**   **Typical message:** This instruction shall not have a TBAA access tag!.

Only memory-accessing instructions may legally carry TBAA tags. Attaching !tbaa metadata to other instructions (such as ret) violates verifier rules.

```
define void @r9() {
entry:
  ; 'ret' is not a memory access and must not carry a TBAA tag.
  ret void, !tbaa !0
}

!0 = !{ !"tbaa_root" }
```

**(R10) Malformed shufflevector mask.**   **Typical message:** shuffle mask element must be in range [-1, N).

The shuffle mask must be a vector of integers in the valid index range for the concatenated inputs. Out-of-range indices or wrong element counts trigger verifier errors.

```
define <4 x i32> @r10(<4 x i32> %v1, <4 x i32> %v2) {
entry:
  ; 8 and 9 are out of range for two 4-element input vectors.
  %mask = <i32 0, i32 1, i32 8, i32 9>
  %x = shufflevector <4 x i32> %v1, <4 x i32> %v2, <4 x i32> %mask
  ret <4 x i32> %x
}
```

**(R11) Invalid getelementptr indices for the pointee type.**   **Typical message:** error: invalid getelementptr indices.

The element type and index sequence of a getelementptr must be compatible with the pointee type of the base pointer.

```
@a = external dso_local global [100 x i32], align 16

define i32 @r11(i64 %idx) {
entry:
```

```
  ; Treat @a as if it were a pointer to i64, which mismatches its real type.
  %p = getelementptr i64, ptr @a, i64 0, i64 %idx
  %x = load i32, ptr %p
  ret i32 %x
}
```

**(R12) Illegal self-reference outside PHI nodes.** **Typical message:** `Only PHI nodes may reference their own value!`.

Only PHI nodes are allowed to refer to their own SSA name as an operand; other instructions may not mention their own result on the right-hand side.

```
define i32 @r12() {
entry:
  ; %x uses itself as an operand in its own definition, which is illegal.
  %x = add i32 %x, 1
  ret i32 %x
}
```

**(R13) PHI entries do not match CFG predecessors.** **Typical message:** `PHI node entries do not match predecessors!`.

The list of incoming blocks for a PHI node must match the set of predecessors of its parent basic block.

```
define i32 @r13(i1 %cond) {
entry:
  br i1 %cond, label %then, label %else

then:
  br label %merge

else:
  br label %merge

merge:
  ; %merge's predecessors are %then and %else, but the PHI mentions %entry.
  %x = phi i32 [ 0, %then ], [ 1, %entry ]
  ret i32 %x
}
```

**(R14) Missing PHI entry for a predecessor.** **Typical message:** `PHINode should have one entry for each predecessor of its parent basic block!`.

Even if all incoming blocks are valid predecessors, the PHI must have exactly one incoming value per predecessor.

```
define i32 @r14(i1 %cond) {
entry:
  br i1 %cond, label %then, label %else

then:
  br label %merge

else:
  br label %merge

other:
  br label %merge

merge:
  ; %merge has three predecessors (%then, %else, %other),
  ; but the PHI only has two entries and omits the %else predecessor.
  %x = phi i32 [ 0, %then ], [ 1, %other ]
  ret i32 %x
}
```

**(R15) Definition does not dominate all uses.** **Typical message:** `Instruction does not dominate all uses!`.

A value must be defined on all paths that reach any of its uses.

```llvm
define i32 @r15(i1 %cond) {
entry:
  br i1 %cond, label %then, label %merge

then:
  %x = add i32 1, 2
  br label %merge

merge:
  ; %x is defined only along the %then path, but used in %merge.
  %y = add i32 %x, 1
  ret i32 %y
}
```

**(R16) Multiple PHI entries for the same predecessor.** **Typical message:** `PHI node has multiple entries for the same basic block with different incoming values!`.

Each predecessor may appear at most once in the PHI incoming list.

```llvm
define i32 @r16(i1 %cond) {
entry:
  br i1 %cond, label %loop, label %exit

loop:
  br label %exit

exit:
  ; The PHI has two entries both coming from %entry, with different values.
  %x = phi i32 [ 32, %entry ], [ 0, %entry ]
  ret i32 %x
}
```

**(R17) PHI nodes not grouped at the top of the basic block.** **Typical message:** `PHI nodes not grouped at top of basic block!`.

All PHI nodes in a basic block must appear before any non-PHI instructions in that block.

```llvm
define i32 @r17(i32 %a, i32 %b) {
entry:
  br label %merge

merge:
  %tmp = add i32 %a, %b      ; non-PHI appears before the PHI
  %x   = phi i32 [ %a, %entry ], [ %b, %entry ]
  ret i32 %tmp
}
```

**(R18) Parser error: expected type token.** **Typical message:** `error: expected type`.

This signature covers syntax errors where the parser expects a type token but does not find a valid one.

```llvm
define i64 @r18() {
entry:
  %1 = icmp ugt i64 2, 1
  ; 'neg' appears where a type such as 'i1' is expected.
  %2 = zext neg i1 %1 to i64
  ret i64 0
}
```

**(R19) Parser error: expected value token.**   **Typical message:** `error: expected value token.`

Similarly, this pattern covers cases where a value operand is missing or malformed, even though the surrounding instruction form is otherwise valid.

```
define i32 @r19() {
entry:
  ; Missing first operand after 'i32': the parser expects a value token here.
  %x = add i32, 1
  ret i32 %x
}
```

**(R20) Use of undefined SSA value.**   **Typical message:** `error: use of undefined value.`

An SSA name is referenced on the right-hand side of an instruction without any dominating definition in the function.

```
define i32 @r20() {
entry:
  ; %x is never defined anywhere in the function.
  %y = add i32 %x, 1          ; use of undefined value %x
  ret i32 %y
}
```

**(R21) Use of undefined metadata.**   **Typical message:** `error: use of undefined metadata.`

Metadata references (e.g., debug info or TBAA tags) must point to defined metadata nodes. Referring to an undefined metadata ID is rejected by the verifier.

```
define void @r21(i32* %p) !dbg !42 {
entry:
  ; !42 is referenced as debug metadata but never defined.
  %x = load i32, i32* %p, !dbg !42
  ret void
}
```

These 21 error signatures cover the dominant verifier and parser failures we observe in LLM-generated LLVM IR. By targeting them explicitly, the REPAIR track encourages methods that can proactively avoid or automatically fix the structural and typing violations that currently prevent such IR from entering production compiler pipelines.

### E.2. Error Injection Procedure

We construct REPAIR instances by injecting the above error patterns into verifier-clean IR, following a deterministic yet flexible procedure.

**Source pool.**   We start from the same pool of LLVM IR functions used in the other tracks (IR-Optset, LLVM test suite, LFK, TSVC, PolyBench, and hand-written kernels; see Appendix H. All source functions pass the LLVM verifier and Alive2 equivalence checks against their original high-level code under our harness.

**Selecting error types and locations.**   For each clean function we sample one to three error types from the taxonomy (R1–R21). We design the sampling process so that all error categories occur at roughly uniform frequency in the final dataset. Given a chosen error type, we select a location in the function that satisfies its preconditions (for example, an existing PHI node, a candidate `getelementptr`, or a load/store with metadata).

**Applying structured edits.**   Each error type is implemented as a structured source-to-source transformation on IR: we mutate opcodes, types, indices, PHI incoming lists, or metadata attachments in a way that deterministically triggers the target verifier or parser message. The rest of the function is left untouched. This contrasts with naive character-level corruption and results in more realistic, compiler-like failures.

**Ensuring repairability.** For every corrupted instance we verify that there exists at least one semantics-preserving repair. Concretely, we run a minimal-fix script that applies a small set of IR transformations tailored to each error type (e.g., restoring a missing PHI entry, fixing a type token, adding a definition), then re-run the verifier and equivalence checker against the original function. Instances for which no such repair can be found are discarded. This guarantees that REPAIR is a well-posed task: models are not asked to fix inherently ambiguous or unrecoverable programs.

## F. Transform Track Details

### F.1. Instance Families and Identifiers

The TRANSFORM track is organized into five families of cases, each containing 60 instances. In CIRBench, their identifiers follow the pattern `T001_Loops_001`, `T002_Ops_001`, `T003_Code_001`, `T004_Module_001`, `T005_Challenge_001`, and so on. Below we summarize the motivation and design of each family.

1. **T001_Loops.** Loops are among the most common and performance-critical structures in real-world programs, especially in numerical and HPC workloads. A large body of work has focused on loop optimizations (e.g., tiling, unrolling, vectorization, interchange), often achieving speedups beyond stock compiler pipelines. Recent LLM-for-compilers systems also target loop transformation at the IR level (Wu et al., 2022; Cummins et al., 2021; Taneja et al., 2025; Zheng et al., 2024). Since even a small change in unroll factors or vectorization patterns can significantly impact runtime, we include a substantial number of loop-centric cases to probe whether models can generate semantically correct, performance-aware loop transformations.

2. **T002_Ops.** In modern ML systems, *operators* (micro-kernels, fused compute kernels, etc.) are the main target of both automatic and manual tuning. Many production stacks lower high-level operators into LLVM IR and then optimize at that boundary. The `T002_Ops` family collects representative operator-style IR kernels in order to stress-test models on fine-grained, numerically delicate transformations that resemble real deployment targets.

3. **T003_Code.** The `T003_Code` family consists of *algorithmic* and *contest-style* code fragments. These examples are typically irregular, branch-heavy, and use problem-specific idioms that challenge traditional optimization heuristics. We include competitive-programming–style and textbook algorithmic snippets to examine whether models can still propose sound transformations when the structure deviates from textbook kernels and dense linear algebra.

4. **T004_Module.** `T004_Module` isolates module-level cases, where a small pipeline spans multiple functions (and sometimes multiple translation units) with shared globals and helper utilities. Most existing LLM-for-IR work focuses on function-local transformation; module-level reasoning introduces additional complexity via inter-procedural effects and global state. In our experiments, models systematically perform worse on module-level cases than on single-function kernels, so we include a deliberately challenging module family to evaluate inter-procedural reasoning and to surface failure modes.

5. **T005_Challenge.** `T005_Challenge` contains cases with known *super-optimization potential*: for each instance we can construct an IR variant that is strictly faster than the baseline compiler's `-O3` output while remaining semantically equivalent under our protocol. These examples are selected so that the improvement is both measurable and well-understood. The goal of this family is to stress-test whether models (and downstream pipelines) can detect such latent opportunities and synthesize IR that surpasses the standard optimization pipeline, while still satisfying layered correctness constraints.

### F.2. T001 Loops and T005 Challenge: External Sources and Mining

Beyond our in-house examples, the TRANSFORM track leverages established benchmarks and existing code corpora.

**T001_Loops.** The `T001_Loops` cases are drawn from three classical loop-oriented suites: LFK (McMahon, 1986), TSVC (Taneja et al., 2025), and PolyBench (Pouchet et al., 2016). LFK and TSVC provide fine-grained loop kernels that stress alias analysis, dependence testing, and vectorization legality, while PolyBench offers structured stencil and linear-algebraic kernels with well-defined input/output behavior. From these suites we extract and normalize single-loop or small nest fragments where the semantics are clear enough for equivalence checking. We keep instances where non-trivial unrolling, vectorization, or loop-reordering opportunities exist and filter out cases that are trivially optimized by stock `-O3` or dominated by undefined behavior.

**T005_Challenge.**    The `T005_Challenge` cases are mined from a large C/C++ code corpus similar to CodeNet. We run a multi-stage search on top of a fixed `-O3` baseline:

- profile-guided optimization (PGO) with representative inputs,

- an LLVM-based autotuner that explores alternative pass sequences and flags, and

- multiple rounds of `-O3`-like pipelines with small variations (e.g., unrolling thresholds, vectorization hints).

For each candidate function we keep only those where we can (i) find at least one IR variant that consistently outperforms the `-O3` baseline by $\geq 5\%$ on our reference platform and (ii) validate semantic equivalence using Alive2 and our checksum-based harness. These become our "super-optimization witnesses": each case admits at least one IR that is strictly better than what the stock compiler emits at `-O3`. As emphasized in the main text, this family is a *stress-test* for high-headroom patterns and is not intended to reflect prevalence in typical workloads.

### F.3. T002_Ops: Operator-Style IR Kernels

The `T002_Ops` family consists of operator-style kernels that mirror the building blocks of modern ML and signal-processing workloads. We organize them into several sub-groups and instantiate each with multiple shapes, strides, and datatype combinations.

**Elementwise and activation kernels.**    This group includes smooth activations such as SiLU/Swish ($y = x \cdot \sigma(x)$ with numerically stable sigmoid), GELU with tanh-based approximation, Mish ($x \cdot \tanh(\text{softplus}(x))$), and Softplus with branch-free implementations. We also cover Softsign, Hard-Swish (implemented via `min`/`max` clamps), LeakyReLU, and PReLU with per-channel parameters. These kernels combine elementary operations (add/mul/exp/log/tanh) in ways that expose algebraic simplification, FMA fusion, and branch elimination opportunities, while stressing the model's ability to preserve corner-case semantics (overflow, saturation, stability).

**Normalization and mean–variance reduction.**    We include RMSNorm, LayerNorm, BatchNorm (inference mode), and GroupNorm. All involve reductions over one or more dimensions followed by affine rescaling with learned parameters. The IR expresses mean/variance computation, $\epsilon$-stabilized square roots, and broadcasting of $\gamma$ and $\beta$. These cases test whether models can safely restructure reduction-and-broadcast patterns and reason about normalization axes.

**Reductions and softmax family.**    We include 1D Softmax and LogSoftmax (via numerically stable log-sum-exp), L2 normalization, and small-$K$ Top-$K$ selection. Softmax-style kernels require subtracting the maximum for stability and chaining `exp`, reduction, and division; Top-$K$ admits multiple algorithmic strategies (partial sorting vs. heap-based selection). These examples exercise both numerical robustness and control-flow restructuring.

**Pooling and light convolution.**    We provide MaxPool2D and AvgPool2D with $2 \times 2$ kernels and stride 2, as well as depthwise Conv2D with $3 \times 3$ kernels. They involve sliding-window access patterns, boundary handling, and opportunities for loop unrolling and vectorization that are sensitive to alignment and padding assumptions.

**Lightweight tensor algebra and broadcasting.**    This group includes GLU (gated linear unit), Add+LayerNorm fusion, simple Scale–Shift ($y = x \cdot \gamma + \beta$) with varied strides, clipping with well-defined NaN behavior, and small piecewise losses. These stress broadcasting rules, fusion patterns, and branch-free implementations of piecewise functions.

**Deterministic loss and probability kernels.**    We add Huber and SmoothL1 losses and CrossEntropy composed with LogSoftmax for sparse labels. All are deterministic (no RNG), but feature piecewise definitions and numerically delicate regions, testing whether models can refactor loss computations while preserving stability and avoiding spurious branches.

**Quantization and low-precision arithmetic.**    Finally, we include per-tensor affine quantization (u8), dequantization, and requantization (u8→u8), alongside channel shuffle and bilinear resize operators. Quantization kernels require precise integer rounding, clamping, and scaling semantics that must be preserved across architectures. These examples explicitly probe whether IR rewrites respect bit-exact behavior and avoid introducing undefined shifts or overflow.

## F.4. T003_Code: Algorithmic and Contest-Style Snippets

The T003_Code family contains algorithmic kernels inspired by competitive-programming and textbook problems. Unlike the operator kernels above, these snippets are control-heavy, often branchy, and rely on intricate data-structure invariants. We group them by algorithmic pattern rather than by application domain.

**Sliding windows, two-pointers, and monotone structures.** We include sliding-window maximum (monotone deque), shortest-subarray variants with prefix sums and monotone queues, window-counting tasks, and pair-distance problems solved via binary search on the answer. These examples stress loop invariants, amortized data-structure costs, and non-trivial exit conditions.

**String algorithms.** We cover suffix arrays with Kasai's LCP, suffix automata, Aho–Corasick automata, Manacher's algorithm, Z-algorithms, rolling hashes, and KMP-based periodicity/border computations. Such code features tight loops over arrays of characters and indices, multiple nested invariants, and condition-heavy transitions. Transformations must not break automaton invariants or indexing arithmetic.

**Graphs, flows, and connectivity.** We include Dinic-style max-flow, min-cost max-flow, Hopcroft–Karp bipartite matching, 2-SAT via SCC, bridge and articulation-point detection, and optimized shortest-path variants (0–1 BFS, bucketed Dijkstra). These kernels involve complex graph traversals and mutable adjacency structures, testing robustness on irregular control flow and pointer-rich IR.

**Trees and tree-based algorithms.** We add tree LCA (binary lifting), heavy-light decomposition with segment trees, classic tree DP (independent set/vertex cover), rerooting DP, DSU-on-tree, and k-th ancestor queries. These examples stress recursion-to-iteration transformations, stack discipline, and layout-sensitive pointer arithmetic on tree representations.

**Dynamic programming and optimization techniques.** We cover LIS in $O(n \log n)$, interval DP, divide-and-conquer–optimized DP, Convex Hull Trick and Li Chao trees, digit DP, and bitmask DP on DAGs or TSP-like problems. Such code intentionally mixes algorithmic insights with low-level micro-optimizations (loop ordering, cache-friendly layouts), making it a challenging yet realistic target for IR-level rewriting.

**Computational geometry.** We include convex hull construction, closest-pair-of-points via divide-and-conquer, rotating-calipers routines, sweep-line segment-intersection detection, and point-in-polygon tests. These kernels feature geometric predicates and delicate degeneracy handling; any transformation must preserve orientation tests, tolerance thresholds, and early-exit conditions.

**Number theory and algebra.** We include 64-bit factorization (Miller–Rabin plus Pollard Rho), NTT-based convolution, extended Euclid and CRT, linear sieves and multiplicative-function precomputation, and linear recurrences via matrix exponentiation or Kitamasa-like methods. These snippets rely on modular arithmetic identities and overflow-sensitive integer operations, stressing correctness of low-level arithmetic rewrites.

**Advanced data structures and "black magic" techniques.** We provide persistent segment trees, DSU with rollback, 2D Fenwick/segment trees, block-based decompositions, and sparse tables for RMQ. We also cover Mo's algorithm, CDQ divide-and-conquer, parallel binary search, and meet-in-the-middle patterns. These examples combine non-trivial ordering of queries/updates with careful control over memory footprints and cache reuse, ensuring that CIRBench is not limited to regular kernel-style code.

## F.5. T004_Module: Module-Level Pipelines

The T004_Module family contains module-level examples that span multiple functions and, in many cases, multiple translation units. Each case forms a small pipeline with several stages, constants, and helper functions that can be specialized or inlined. Our goal is to test inter-procedural reasoning, attribute propagation, and module-level optimization behavior.

**Image and vision pipelines.** We include pipelines such as Bayer demosaicing, YUV→RGB conversion with gamma LUTs, histogram equalization, integral-image–based box filters, morphology (erode/dilate), Harris corner detection, image

pyramids, bilateral filtering, seam carving, and connected-component labeling. Typical patterns include fixed kernel sizes, read-only coefficient tables, and helper functions that are good candidates for cloning, specialization, and cross-TU inlining.

**Audio, video, and codec fragments.** We construct audio/video kernels such as polyphase resampling, cascaded biquad equalizers, MDCT with windowing, deblocking filters, motion-estimation inner loops, reversible colour transforms, simple denoisers, and Viterbi decoders. Pipelines are decomposed into small helpers (e.g., tap application, windowing, metric updates) with parameters exposed as constants. These examples exercise readonly/noalias attribute inference, global constant propagation, and profitable inlining across multiple stages.

**Parsing, text processing, and Unicode.** We add module-level pipelines for URL parsing, HTTP header normalization, Base64 encoding with SIMD and scalar fallbacks, regex-style scanners, edit-distance computation, Unicode normalization, and grapheme-boundary segmentation. They employ dispatch tables, configurable parsing modes, and small combinators passed via function pointers. Optimization opportunities include indirect-call promotion, mode specialization, and selective inlining of hot-path helpers while keeping cold paths intact.

**Cryptography and compression.** This group covers ChaCha20+Poly1305-style AEAD, hash functions, modular arithmetic kernels, rANS or arithmetic coding cores, and LZ-family match-finders and encoders. Many parameters (block sizes, fanout, code parameters) are constants, and core round/update routines are small enough to inline. These modules probe whether models can expose constant folding and specialization opportunities without violating strict bit-level semantics.

**Numeric/HPC/graph pipelines.** We include sparse/dense conjugate gradient solvers, Poisson stencils with iterative relaxations, multigrid V-cycles, prefix scans, PageRank-like iterations, and KD-tree builds with queries. These pipelines combine sparse/dense linear algebra with graph traversals and tree-based search, testing module-level alias analysis and inter-procedural constant propagation.

**Data systems, OS, and networking.** Finally, we provide module-level cases from data systems and networking: hash-based and sort-merge joins, group-by aggregations, RLE+bitpacking encoders, Bloom filters, cache-replacement policies, protocol-header parsing (Ethernet/IP/TCP), and log-processing pipelines that use callbacks for filters and projections. These cases feature mixed control/data patterns and abundant opportunities for indirect-call promotion, schema-driven specialization, and inlining of hot callbacks.

Across all `T004_Module` cases, we ensure that the module structure is rich enough that purely local (single-function) reasoning is insufficient, while keeping the overall size manageable for equivalence checking and the layered correctness protocol.

### F.6. Runtime Normalization and Harness

To obtain stable yet affordable timing measurements across kernels, we normalize runtime using a common C harness. Each Transform instance is compiled into a standalone binary whose `main` function is generated from a small macro library (see repository for full source).

**Input sizes and outer loops.** For each kernel we define three representative input scales (e.g., problem sizes $N_1$, $N_2$, $N_3$) and an *outer-loop level* $\ell \in \{0, \ldots, 8\}$ controlling the number of repetitions. Default repetition counts for each level are chosen so that, on our reference AMD Ryzen 9 7950X (Zen4) system, total runtime for the middle scale ($N_2$, $\ell = 2$) typically lies in the 0.5–10s range. Repetition counts can be overridden via environment variables (`BENCH_OUTER0`–`BENCH_OUTER8`) to adapt to other machines.

**Deterministic initialization.** Inputs are initialized with inexpensive XorShift-based pseudo-random generators seeded by a deterministic case identifier. This ensures that different runs and different IR variants see identical inputs, enabling deterministic checksums and fair speedup comparisons.

**Timing.** We measure wall-clock time using `clock_gettime(CLOCK_MONOTONIC)` when available, falling back to `gettimeofday` otherwise. The harness reports a single elapsed time per run along with a checksum and loop count in a structured format: `kernel,KID,caseX,Ll,checksum=0x...,time=...,loops=...`. For each IR variant we execute the binary for a fixed number of repetitions $R$ and average times across runs before computing speedup relative to the `-O3` baseline.

**Tolerant hashing for numeric outputs.** For floating-point outputs we use a tolerant hashing scheme to account for benign rounding differences across hardware and compilers. Given an array of doubles, we canonicalize each value by masking a small number of mantissa bits and collapsing tiny values and NaNs to canonical representatives, then compute an FNV-1a hash of the resulting bit patterns. Tolerance parameters (number of masked mantissa bits and absolute cutoff for zero) have conservative defaults and can be overridden via environment variables if needed. Integer and byte outputs use plain FNV-1a hashing without tolerance. Two runs are treated as semantically equivalent at the harness level if their checksums match.

### F.7. Static and Dynamic Tooling

**Static performance via `llvm-mca`.** For each candidate IR we invoke `llvm-mca` with the same target triple and microarchitecture as used for code generation (Zen4 back-end). We feed the kernel's inner loop and report per-iteration estimates such as throughput and critical-path length. In the main text we summarize these as static speedup factors over the `-O3` reference; the exact `llvm-mca` command lines are provided in the released scripts.

**Dynamic performance via `perf`.** Dynamic measurements use the harness described above. On Linux we obtain wall-clock time either from the harness output itself or, optionally, through `perf stat` wrappers. All kernels run single-threaded with fixed CPU frequency and pinned cores. For each instance, we execute the kernel 10 times and average the runtimes to reduce run-to-run variability. Speedups in Table 5 and Figure 6 are computed from these per-instance averaged runtimes, and we report medians and ranges together with 95% bootstrap confidence intervals as described in Appendix L.

### F.8. Formal vs. test-based equivalence

CIRBench reports **Equiv@$k$** as a best-available semantic guard, following practical compiler-style validation at the IR boundary. For single-function instances, we run Alive2 and take its verdict whenever it is *conclusive*. For module-level instances, and for single-function cases where Alive2 is *inconclusive* (e.g., `timeout/unknown/unsupported/out-of-memory`), we validate candidates using a checksum-based harness. The harness executes a small suite of deterministic stress tests with varying input sizes/iteration budgets and compares observable outputs between the candidate and `ref`. For floating-point outputs, we compare with a fixed tolerance and hash the normalized results. We emphasize that checksum agreement is *test-based* rather than a proof, and its strength depends on test coverage; nevertheless, checksum-style validation is standard in kernel-style benchmark suites where outputs are large and correctness must be checked cheaply at scale (e.g., TSVC). (Maleki et al., 2011)

To make the limitations of bounded translation validation explicit, we additionally report a conservative lower bound, **Equiv_formal@$k$**, on single-function Transform instances. A candidate is credited only if Alive2 *proves* equivalence; Alive2 outcomes {`not-eq`, `timeout`, `unknown`, `unsupported`} are counted as 0. Alive2 is a bounded translation-validation tool, so inconclusive outcomes can arise from solver/resource limits rather than incorrect transformations (Lopes et al., 2021). As a result, Equiv@$k$ can exceed Equiv_formal@$k$; we interpret the gap primarily as a *coverage/decidability* effect of bounded validation, rather than as evidence of systematic mismeasurement. Importantly, when Alive2 returns `not-eq`, we do *not* override it with harness agreement; checksum validation is used only when Alive2 is inconclusive.

*Table 8.* Equiv_formal@$k$ on single-function Transform instances (credited only when Alive2 *proves* equivalence).

| Model | Equiv_formal@1 (%) | Equiv_formal@5 (%) |
|---|---|---|
| GPT-5 | 40.8 | 48.3 |
| Claude Sonnet 4.5 | 12.1 | 21.7 |
| Gemini 2.5 Flash | 28.8 | 38.8 |
| Grok 4 Fast | 17.9 | 39.2 |
| Qwen3-Max | 21.3 | 42.1 |
| Deepseek-V3.2-Exp | 12.9 | 28.8 |

## G. Difficulty thresholds

As discussed in §3.1, we stratify instances in each track into Easy/Medium/Hard (E/M/H) buckets. Our goal is not to define a sophisticated difficulty metric, but to use simple, transparent IR-level features so that difficulty assignments are easy to inspect and reproduce. This appendix describes how we assign E/M/H labels in each track and summarizes the resulting

ranges.

Across tracks we follow a common pattern: we compute a few basic features for each instance (e.g., IR length, control-flow size, number of injected errors), inspect their empirical distributions, and choose static thresholds that split instances into three coarse regimes (short/simple, medium, long/complex). We then script the assignment of E/M/H using these thresholds. The concrete ranges for each track are reported in Table 9.

## G.1. Analysis

Each Analysis instance corresponds to a single LLVM IR function and a set of questions (A1–A5) about that function. We characterize the function by two quantities:

- IR length, measured as the number of LLVM instructions;

- JSON complexity, measured as the number of key–value pairs (for A1) or the number of graph nodes/edges serialized into the answer (for A2–A4).

We then proceed in two steps.

First, we use a simple script to compute histograms of IR length and JSON complexity over the 100 functions. Based on these histograms, we manually choose two length thresholds $L_{\text{short}} < L_{\text{long}}$ and two JSON thresholds $C_{\text{simple}} < C_{\text{complex}}$ such that: short functions (length $\leq L_{\text{short}}$) cover roughly the lower third of the length distribution, long functions (length $\geq L_{\text{long}}$) cover the upper third, and JSON complexity is similarly split into simple / moderate / complex regimes. The exact thresholds are listed in Table 9.

Second, we assign difficulty labels according to the combination of length and JSON complexity: short-and-simple instances are marked Easy; long-and-complex instances are marked Hard; the remaining cross-regime combinations (e.g., short-but-complex or medium length) are marked Medium. The same E/M/H label is shared across all subtasks A1–A5 for a given function.

## G.2. Refactor

Refactor instances are also single functions. For each reference (post-pass) IR we compute the total number of LLVM instructions and sort all 200 instances by length. We then use length-only thresholds to assign difficulty:

- Easy: functions in the shortest third of the length distribution;

- Medium: functions in the middle third;

- Hard: functions in the longest third.

The precise ranges in terms of instruction counts are given in Table 9. Refactor-Normal and Refactor-Reverse share the same E/M/H labels.

## G.3. Repair

Repair instances are constructed by injecting 1–3 structured errors into verifier-clean functions. Here difficulty depends both on IR size and on how many errors must be fixed.

We first assign a coarse E/M/H label based on IR length alone, using tertile-style cut points computed on the Repair subset. Due to the way Repair instances are generated and filtered, IR lengths are more clustered than in Refactor, which yields narrower empirical spans. Within each length band we then further distinguish instances by the number of injected errors $e \in \{1, 2, 3\}$: single-error cases are treated as easier than multi-error ones. In the main text we report difficulty at the E/M/H granularity; Figure 4 additionally breaks down results by (E1/E2/E3), where Ex denotes $x$ injected errors regardless of length.

Concretely, we map (length band, error count) to E/M/H via a simple rule-based script: shorter instances with fewer injected errors are labeled Easy, larger instances with more errors are labeled Hard, and the remaining combinations are labeled Medium. The resulting length ranges are summarized in Table 9.

## G.4. Transform

Transform instances are kernel-sized functions or small modules. For each instance we use the IR length of the unoptimized (near `-O0`) version as the primary difficulty signal:

- we compute the number of LLVM instructions in the `-O0` IR;

- we sort all 300 instances by this length;

- we choose two cut points that roughly split the distribution into three equal-sized groups.

Instances in the lowest, middle, and highest length groups are labeled Easy, Medium, and Hard respectively. We do not introduce a separate difficulty label for the Challenge ("Super") subset; by construction, most of its programs fall into the upper part of the length distribution and are therefore assigned Hard.

## G.5. Summary of ranges

Table 9 summarizes the empirical IR length ranges for Easy/Medium/Hard instances in each track. These ranges are descriptive rather than prescriptive: all difficulty labels are assigned by the simple rules above, and the table reports the resulting spans.

*Table 9.* Empirical instruction-count ranges for Easy/Medium/Hard instances in each track. Ranges are inclusive and reported in terms of LLVM instruction counts on the relevant IR (function-level for Analysis/Refactor/Repair, near-`-O0` IR for Transform). Repair ranges may appear narrower because the generated instances are length-clustered after filtering.

| Track | Easy | Medium | Hard |
|---|---|---|---|
| Analysis | *[8, 149]* | *[9, 187]* | *[11, 588]* |
| Refactor | *[63, 95]* | *[65, 226]* | *[144, 295]* |
| Repair | *[40, 43]* | *[100, 103]* | *[200, 219]* |
| Transform | *[133, 1428]* | *[236, 3840]* | *[414, 5347]* |

# H. Toolchain

This section describes the concrete toolchain we use to build and run CIRBench, as well as the preprocessing and curation steps that turn raw programs into benchmark instances. All scripts referenced below are included in the public repository.

## H.1. Environment and dependencies

We run all experiments on Ubuntu 22.04.4 LTS with LLVM 19.1.0 and Alive2 v19.0. The following commands create a local Python environment, build LLVM/Clang and Alive2, and install the CIRBench driver:

```
CIRBENCH_HOME=$(pwd)

# Python env and cirbench package
bash env/setup_env.sh
source .venv/bin/activate
pip install -e .

# LLVM/Clang (llvmorg-19.1.0)
git clone --depth=1 --branch llvmorg-19.1.0 \
  https://github.com/llvm/llvm-project.git
cd $CIRBENCH_HOME/llvm-project
mkdir build && cd build
cmake -G "Unix Makefiles" \
  -DCMAKE_BUILD_TYPE=Release \
  -DLLVM_ENABLE_PROJECTS="clang" \
  -DLLVM_TARGETS_TO_BUILD="host" \
  -DLLVM_ENABLE_RTTI=ON \
  -DLLVM_ENABLE_EH=ON \
  ../llvm
```

```
make -j$(nproc)

# Alive2 (v19.0) built against the same LLVM
cd $CIRBENCH_HOME/alive2
git checkout v19.0 -f
mkdir build && cd build
cmake -G "Unix Makefiles" \
  -DCMAKE_BUILD_TYPE=Release \
  -DBUILD_TV=1 \
  -DCMAKE_PREFIX_PATH=$CIRBENCH_HOME/llvm-project/build ..
make -j$(nproc)

cd $CIRBENCH_HOME
```

### H.2. IR generation, slicing, and normalization

We compile all source benchmarks (IR-Optset, LLVM tests, LFK, TSVC, PolyBench, and hand-written kernels) with `clang` at `-O0` to obtain unoptimized LLVM IR. A custom Python toolchain then performs:

1. **Slicing.** We extract primarily function-level IR fragments, plus a small number of module-level kernels that retain all in-scope callees and globals. Each slice is required to compile to a runnable binary under the common harness.

2. **Normalization.** We normalize SSA value names, basic-block labels, and whitespace, and strip non-semantic debug metadata. This makes IR comparisons (EM and deduplication) robust to cosmetic differences while preserving all semantics-relevant structure.

3. **Oracle computation.** For each slice we compute Analysis labels using LLVM's built-in analyses and auxiliary scripts, Refactor reference outputs using single-pass `opt` invocations, Repair targets via the minimal-fix script, and Transform performance oracles (static and dynamic) using the harness described in Appendix F.

### H.3. Instance curation, UB filtering, and deduplication

Not all IR fragments are suitable as benchmark instances. We therefore apply several curation steps before packaging:

- **Exact and near-duplicate removal.** After normalization we compute hashes over the textual IR of each slice and drop exact duplicates across and within source suites. In addition, a simple feature-based script (opcode histograms and basic block counts) flags near-duplicates; we keep at most one representative per cluster. This avoids overweighting any single kernel or micro-benchmark.

- **UB-dominated examples.** Some suites contain tests whose primary purpose is to exercise undefined behavior (e.g., sanitizer regression tests). We automatically filter obvious UB stress tests based on directory patterns and instruction kinds, and manually inspect the remaining candidates. IR fragments whose intended behavior is dominated by UB (for example, where most dynamic paths immediately trigger UB under the harness) are discarded.

- **Timing stability for Transform.** For Transform kernels we sanity-check runtime stability by compiling the reference `-O3` version, running it repeatedly under the fixed harness and hardware (Appendix C), and computing the coefficient of variation. Instances with highly unstable timings under these controlled settings are removed.

Together, these steps realise the curation statement in subsection 3.3: we remove exact and near-duplicate IR fragments and filter out cases that are UB-dominated or exhibit unstable timing behavior. The scripts implementing these checks are provided under `tools/` in the repository and can be re-run or customised by users.

### H.4. Running CIRBench via the CLI

The `cirbench` command-line interface wraps the toolchain and provides a uniform way to audit, list, and run tasks. Assuming the environment above is configured and CIRBENCH_HOME points to the repository root, a typical workflow is:

**Sanity checks and listing.**

```
# audit the environment and configuration
cirbench doctor --cfg configs/cirbench.yaml

# list available instances per track
cirbench list --task analysis
cirbench list --task transform
cirbench list --task repair
cirbench list --task refactor
```

**Single-instance runs.** We use an ISO-8601 timestamp as a run identifier:

```
export CIRBENCH_RUN_ID=$(date -u +"%Y-%m-%dT%H-%M-%SZ")
export GEMINI_API_KEY=...   # or other model-specific keys

# Analysis
cirbench run --task analysis \
  --model gemini:gemini-2.5-flash \
  --cfg configs/cirbench.yaml \
  --debug --concurrency 1 \
  --select "A001_alias_001"

# Repair (Hint / Hard)
cirbench run --task repair \
  --model gemini:gemini-2.5-flash \
  --cfg configs/cirbench.yaml \
  --repair-mode normal \
  --debug --concurrency 1 \
  --select "Repair_001"

cirbench run --task repair \
  --model gemini:gemini-2.5-flash \
  --cfg configs/cirbench.yaml \
  --repair-mode hard \
  --debug --concurrency 1 \
  --select "Repair_001"

# Refactor (Normal / Reverse)
cirbench run --task refactor \
  --model gemini:gemini-2.5-flash \
  --cfg configs/cirbench.yaml \
  --refactor-mode normal \
  --debug --concurrency 1 \
  --select "RF003_EarlyCSE_001"

cirbench run --task refactor \
  --model gemini:gemini-2.5-flash \
  --cfg configs/cirbench.yaml \
  --refactor-mode reverse \
  --debug --concurrency 1 \
  --select "RF003_EarlyCSE_001"

# Transform (Direct / Copilot)
cirbench run --task transform \
  --model gemini:gemini-2.5-flash \
  --cfg configs/cirbench.yaml \
  --transform-mode normal \
  --debug --concurrency 1 \
  --select "T001_Loops_001"

cirbench run --task transform \
  --model gemini:gemini-2.5-flash \
  --cfg configs/cirbench.yaml \
  --transform-mode copilot \
```

```
--debug --concurrency 1 \
--select "T001_Loops_001"
```

**Full runs and reporting.** All API-based models were queried between 2025-11-04 and 2025-11-25. To reproduce the experiments in the main paper for a given model, we run:

```
cirbench run --task analysis \
  --model gemini:gemini-2.5-flash \
  --cfg configs/cirbench.yaml

cirbench run --task repair \
  --model gemini:gemini-2.5-flash \
  --cfg configs/cirbench.yaml \
  --repair-mode normal

cirbench run --task repair \
  --model gemini:gemini-2.5-flash \
  --cfg configs/cirbench.yaml \
  --repair-mode hard

cirbench run --task refactor \
  --model gemini:gemini-2.5-flash \
  --cfg configs/cirbench.yaml \
  --refactor-mode normal

cirbench run --task refactor \
  --model gemini:gemini-2.5-flash \
  --cfg configs/cirbench.yaml \
  --refactor-mode reverse

cirbench run --task transform \
  --model gemini:gemini-2.5-flash \
  --cfg configs/cirbench.yaml
```

After the runs complete, we summarise per-run results and aggregate statistics:

```
# per-run JSON/CSV reports
cirbench report --cfg configs/cirbench.yaml \
  --run-id $CIRBENCH_RUN_ID

# aggregate across runs into the tables in the paper
cirbench aggregate --cfg configs/cirbench.yaml \
  --run-id $CIRBENCH_RUN_ID
```

These commands reproduce the evaluation protocol described in subsection 3.3 and section 4, and can be adapted to new models, prompts, or hardware by modifying the configuration file.

## I. Contamination

### I.1. Data sources and timeline

CIRBench draws IR programs from a mix of long-standing benchmark suites (IR-Optset, LLVM test suite, LFK, TSVC, PolyBench, CodeNet) and a small number of hand-written kernels (Section 3.3). Most of these suites were released well before recent code LLMs were trained, and many of the underlying source programs have been available in public repositories for years. At the same time, we do *not* have access to the exact training corpora or cut-off dates of the evaluated models. It is therefore plausible that some of the source programs underlying CIRBench have been seen, directly or indirectly, during pretraining.

In addition, our benchmark operates on LLVM IR rather than source code. This IR is produced by compiling the original suites with a fixed toolchain and then slicing, normalizing, and corrupting it as described in Appendix H. These

transformations change both syntax and structure compared to the original source, but they do not formally rule out pretraining overlap.

### I.2. Within-benchmark deduplication

Our preprocessing removes trivial duplicates *within* CIRBench: IR fragments that are textually identical after normalization, or that differ only in non-semantic metadata, are merged into a single instance before stratifying by difficulty (Appendix H). This step is primarily intended to avoid over-weighting particular patterns in evaluation, rather than to defend against pretraining contamination.

We do not perform any cross-corpus deduplication between CIRBench and the unknown pretraining corpora of the evaluated models. As a result, the benchmark should be interpreted as potentially containing programs that are semantically similar to, or even derived from, training data.

### I.3. Time-based splits

Given the lack of precise information about the models' training cut-off dates and corpora, we do not enforce a strict time-based split between "pre-training" and "post-training" sources. Many of the suites we rely on were created well before modern LLMs, but their code (or close variants) may still have been included in large code crawls.

Instead of attempting a pseudo time-split that would provide a false sense of security, we treat CIRBench as a single held-out test set. All reported results should be read with the understanding that some degree of pretraining overlap is possible, especially for widely used kernels.

### I.4. Limitations and interpretation

We do not conduct a systematic contamination audit, and we cannot quantify the fraction of instances that might be memorized or partially memorized. Consequently, CIRBench does *not* guarantee that all measured behavior reflects purely out-of-distribution generalization.

That said, several aspects of our design reduce the risk that contamination trivially solves the tasks. First, most tracks operate on normalized IR and inject additional structure (e.g., repair errors or specific pass applications), so even if a model saw the original source code, it would still need to reason about the IR-level transformations we apply. Second, our metrics emphasise layered correctness (Valid/Equiv) and performance under strict guards, which are sensitive to small semantic differences and less likely to be saturated by rote memorization.

Overall, we view potential pretraining contamination as an important limitation rather than a solved issue. We hope that future work—for example, using synthetic or proprietary IR sources with known provenance, or collaborating with model providers on joint audits—can further tighten this aspect of CIRBench.

## J. Additional open-weight baselines

### J.1. Models

In addition to the six main models discussed in section 4, we evaluate three further open-weight baselines on CIRBench:

- **gpt-oss-120b**: An open-weight, Mixture-of-Experts (MoE) decoder-only language model from OpenAI, intended for production, high-reasoning general-purpose use. It has about 117B total parameters with 5.1B active per token, fits on a single 80GB GPU (e.g., H100/MI300X), and is trained in the "harmony" response format for code, tools, and agentic workflows.(OpenAI, 2025c;b)

- **Llama 4 Maverick**: A Llama 4 series MoE model from Meta, with roughly 400B total parameters and 17B active parameters per forward pass (128 experts). It is an auto-regressive, multimodal (text+image) instruction-tuned model, optimized for assistant-style dialogue, code generation, and vision-language reasoning across multiple languages.(Meta, 2025; Meta AI, 2025)

- **Qwen3-235B-A22B-Thinking-2507**: A high-capacity, open-weight MoE causal language model from Alibaba's Qwen team, with 235B total parameters and 22B activated experts per token. This "thinking-only" variant is specialized for

complex reasoning, math, science, and long-context tasks (native context up to 262k tokens), and is reported to perform strongly on benchmarks such as AIME, SuperGPQA, LiveCodeBench, and MMLU-Redux (Qwen Team, 2025).

All three models are evaluated with the same prompt templates, decoding hyperparameters, and toolchain as in subsection 4.1 (see Appendix B, Appendix K, and Appendix H). We do not perform any model-specific prompt tuning or parameter sweeps for these baselines; they serve as fully reproducible reference points for future IR-level methods.

### J.2. Results across tracks

*Table 10.* Analysis EM/F1 and Valid@{1,5}, Equiv@{1,5} on the Repair, Refactor, and Transform tracks.

| Model | Analysis | | Repair-Hint | | | | | | Repair-Hard | | | | | | Refactor-Normal | | | | | | Transform | | | |
| | EM | F1 | Valid@ | | Equiv@ | | Valid@ | | Equiv@ | | Valid@ | | Equiv@ | | EM@ | | Valid@ | | Equiv@ | |
| | | | 1 | 5 | 1 | 5 | 1 | 5 | 1 | 5 | 1 | 5 | 1 | 5 | 1 | 5 | 1 | 5 | 1 | 5 |
|---|---|---|---|---|---|---|---|---|---|---|---|---|---|---|---|---|---|---|---|---|
| **gpt-oss-120b** | 47% | 0.704 | 78.5 | 91.0 | 72.0 | 83.0 | 48.0 | 68.0 | 42.0 | 60.5 | 34.5 | 59.0 | 15.5 | 24.5 | 0.0 | 0.0 | 60.9 | 89.5 | 28.2 | 64.7 |
| **llama-4-maverick** | 12% | 0.416 | 55.0 | 76.0 | 47.0 | 68.5 | 46.5 | 70.0 | 37.5 | 59.0 | 35.5 | 71.0 | 16.0 | 27.5 | 0.0 | 0.0 | 42.0 | 69.3 | 12.6 | 27.0 |
| **qwen3-235b-a22b-thinking-2507** | 39% | 0.713 | 74.0 | 89.5 | 66.0 | 82.0 | 36.5 | 61.0 | 33.0 | 56.5 | 22.0 | 47.5 | 8.5 | 16.5 | 0.0 | 0.0 | 52.3 | 82.0 | 0.25 | 0.51 |

*Table 11.* Performance of six models on the Transform task under direct and copilot modes.

| Model | direct | | | | | | copilot | | | | | |
| | mca | | | perf | | | mca | | | perf | | |
| | min | med | max | min | med | max | min | med | max | min | med | max |
|---|---|---|---|---|---|---|---|---|---|---|---|---|
| **gpt-oss-120b** | 0.20 | 0.86↓ | 12.27 | 0.05 | 0.75↓ | 3.32 | 0.38 | 1.18↑ | 6.55 | 0.57 | 0.99↓ | 4.54 |
| **llama-4-maverick** | 0.21 | 0.86↓ | 3.65 | 0.10 | 0.61↓ | 6.38 | 0.97 | 1.19↑ | 6.04 | 0.44 | 0.99↓ | 2.33 |
| **qwen3-235b-a22b-thinking-2507** | 0.22 | 0.93↓ | 12.45 | 0.08 | 0.79↓ | 5.19 | 0.81 | 1.20↑ | 4.38 | 0.15 | 0.99↓ | 4.48 |

Table 10, Table 11 reports the main CIRBench metrics for the three additional open-weight models. Inspecting the numbers, we observe qualitatively similar trends to those in the main text: Analysis EM is consistently higher than IR editing metrics, and the Transform track remains the most challenging axis, with a substantial gap between Valid@1 and Equiv@1 and median speedups clustered near or below parity with -O3. Differences between models are also more pronounced on Repair, Refactor-Normal, and Transform than on Analysis, indicating that most of the variation arises in semantics-preserving IR editing rather than coarse IR understanding.

### J.3. Discussion

These additional results reinforce our main conclusions from section 4: IR understanding is necessary but not sufficient for reliable IR editing; semantics-preserving, performance-aware Transform remains a bottleneck for all model families; and layered correctness metrics (Valid/Equiv) are essential to interpret performance numbers. The open-weight baselines in Table 10 provide fully reproducible anchors for future work that fine-tunes, adapts, or augments LLMs for IR-centric compilation tasks. We relegate them to the appendix for space reasons; they do not change the qualitative conclusions drawn from the six main models in the body of the paper.

## K. Decoding & budgets

### K.1. Decoding hyperparameters

For all generation-style tracks (Refactor-Normal, Repair, Transform) we use the prompt templates in Appendix B and sample $k \in \{1, 5\}$ candidates per instance. Table 12 summarizes the decoding hyperparameters for each model. Temperature and top-$p$ values are chosen from provider-recommended ranges or default settings and are kept fixed across all tracks; we do not tune them on CIRBench. All models share the same stop conditions; maximum generation length is left to the provider defaults. (end-of-sequence token or completion of a function/module). Random seeds are fixed per run to make repeated evaluations deterministic.

### K.2. Token budgets and cost

To make comparisons fair, we match token budgets across models. Table 13 reports, for each model, the total number of calls, average prompt and completion lengths per call, total tokens consumed on CIRBench, and an approximate monetary

*Table 12.* Decoding hyperparameters for all models. Values are kept fixed across tracks.

| Model | Temp. | Top-$p$ | Top-$k$ | Max new toks | $k$ samples | Stop condition | Thinking | Thinking budget |
|---|---|---|---|---|---|---|---|---|
| GPT-5 | 0.1 | 0.9 | - | - | 1, 5 | EOS / `"</IR_OUT>"`, `"</CIR_JSON>"` | default | - |
| Claude Sonnet 4.5 | 0.1 | - | - | - | 1, 5 | EOS / `"</IR_OUT>"`, `"</CIR_JSON>"` | true | 8192 |
| Gemini 2.5 Flash | 0.1 | 0.9 | - | - | 1, 5 | EOS / `"</IR_OUT>"`, `"</CIR_JSON>"` | true | - |
| Grok 4 Fast | 0.1 | 0.9 | - | - | 1, 5 | EOS / `"</IR_OUT>"`, `"</CIR_JSON>"` | default | - |
| Qwen3-Max-Preview | 0.1 | 0.9 | - | - | 1, 5 | EOS / `"</IR_OUT>"`, `"</CIR_JSON>"` | true | 8192 |
| Deepseek-V3.2-Exp | 0.1 | 0.9 | - | - | 1, 5 | EOS / `"</IR_OUT>"`, `"</CIR_JSON>"` | true | 8192 |
| gpt-oss-120b | 0.1 | 0.9 | - | - | 1, 5 | EOS / `"</IR_OUT>"`, `"</CIR_JSON>"` | default | - |
| Llama 4 Maverick | 0.1 | 0.9 | - | 8192 | 1, 5 | EOS / `"</IR_OUT>"`, `"</CIR_JSON>"` | default | - |
| Qwen3-235B-A22B-Thinking-2507 | 0.1 | 0.9 | - | - | 1, 5 | EOS / `"</IR_OUT>"`, `"</CIR_JSON>"` | default | - |

cost based on provider pricing at the time of writing. These numbers are computed over all tracks with $k = 5$ samples, i.e., the worst-case budget; runs with $k = 1$ consume a strict subset of this budget.

*Table 13.* Token usage and approximate cost per model over all CIRBench tracks (with $k = 5$ samples). Prompt/comp. lengths are averaged over calls. Costs are illustrative and should be recomputed with up-to-date pricing.

| Model | #calls | Avg prompt | Avg comp. | Total toks | Total toks (M) | Approx. cost (USD) |
|---|---|---|---|---|---|---|
| GPT-5 | 2689 | 6135 | 7111 | 35618842 | 35.62 | $\approx$ 212$ |
| Claude Sonnet 4.5 | 3915 | 8059 | 4585 | 49505421 | 49.51 | $\approx$ 364$ |
| Gemini 2.5 Flash | 4001 | 8160 | 2556 | 69130608 | 69.13 | $\approx$ 28.5$ |
| Grok 4 Fast | 3738 | 5869 | 6179 | 59308791 | 59.31 | $\approx$ 39$ |
| Qwen3-Max-Preview | 3565 | 7450 | 9180 | 58522141 | 58.52 | $\approx$ 135$ |
| Deepseek-V3.2-Exp | 4505 | 6772 | 8681 | 69481595 | 69.48 | $\approx$ 24$ |
| gpt-oss-120b | 3946 | 6519 | 3471 | 39227433 | 39.23 | - |
| Llama 4 Maverick | 4800 | 6355 | 2568 | 42191879 | 42.19 | - |
| Qwen3-235B-A22B-Thinking-2507 | 4250 | 7888 | 10303 | 76460350 | 76.47 | - |

# L. Statistics / bootstrap CI

## L.1. Bootstrap CIs for EM, Valid@k, and Equiv@k

Most of our metrics are instance-aggregated proportions: exact match (EM) on Analysis, and Valid@$k$/Equiv@$k$ on the Repair, Refactor, and Transform tracks. For a given track, model, and metric, let $\{m_i\}_{i=1}^{N}$ denote the per-instance outcomes (e.g., $m_i \in \{0, 1\}$ for EM or Valid@1, or $m_i \in [0, 1]$ for per-instance Valid@5 / Equiv@5). The reported point estimate is the empirical mean $\hat{m} = \frac{1}{N} \sum_{i=1}^{N} m_i$.

To quantify statistical uncertainty we report non-parametric bootstrap confidence intervals (CIs). For each (model, track, metric) triple we perform $B = 1000$ bootstrap resamples with replacement:

1. For $b = 1, \ldots, B$:

   (a) Sample $N$ indices $i_1, \ldots, i_N$ with replacement from $\{1, \ldots, N\}$.

   (b) Form the bootstrap sample $\{m_{i_1}, \ldots, m_{i_N}\}$.

   (c) Compute the bootstrap mean $\hat{m}^{(b)} = \frac{1}{N} \sum_{j=1}^{N} m_{i_j}$.

2. Let $\hat{m}^{(1)}, \ldots, \hat{m}^{(B)}$ denote the resulting bootstrap distribution. We take the 2.5th and 97.5th percentiles as the lower and upper bounds of a 95% CI: $[\hat{m}^{\text{lo}}, \hat{m}^{\text{hi}}]$.

Table 14, Table 15, Table 16, Table 17 reports the resulting 95% bootstrap CI *half-widths* for the main EM, F1, Valid@$\{1, 5\}$ and Equiv@$\{1, 5\}$ metrics on Analysis, Repair, Refactor-Normal, and Transform. For a given cell with point estimate $p$ and half-width $w$, the corresponding 95% CI is $[p - w, p + w]$. All CIs are computed over the fixed set of instances for each track and do not require any additional model calls beyond the runs used to obtain the point estimates.

*Table 14.* 95% bootstrap CI half-widths (in absolute probability) for EM and F1 on the Analysis track ($B = 1000$ bootstrap resamples). Each entry $w$ corresponds to a CI of the form $[p - w,\ p + w]$ for the point estimate $p$ in the corresponding cell of Table 3.

| Model | EM | F1 |
|---|---|---|
| GPT-5 | $70 \pm 9.00$ | $84.0 \pm 6.03$ |
| Claude Sonnet 4.5 | $59 \pm 10.0$ | $75.3 \pm 7.38$ |
| Gemini 2.5 Flash | $52 \pm 9.50$ | $74.6 \pm 6.89$ |
| Grok 4 Fast | $68 \pm 9.00$ | $82.0 \pm 6.41$ |
| Qwen3-Max-Preview | $52 \pm 9.50$ | $72.2 \pm 7.35$ |
| Deepseek-V3.2-Exp | $41 \pm 9.50$ | $66.4 \pm 7.61$ |
| gpt-oss-120b | $47 \pm 9.50$ | $70.4 \pm 7.39$ |
| Llama 4 Maverick | $12 \pm 6.00$ | $41.6 \pm 7.09$ |
| Qwen3-235B-A22B-Thinking-2507 | $39 \pm 9.00$ | $71.3 \pm 6.29$ |

*Table 15.* 95% bootstrap CI half-widths (in absolute probability) for Valid@$\{1,5\}$ and Equiv@$\{1,5\}$ on the Repair track ($B = 1000$ bootstrap resamples). Each entry $w$ corresponds to a CI of the form $[p - w,\ p + w]$ for the point estimate $p$ in the corresponding cell of Table 4.

| Model | Repair-Hint | | | | Repair-Hard | | | |
|---|---|---|---|---|---|---|---|---|
| | Valid@ | | Equiv@ | | Valid@ | | Equiv@ | |
| | 1 | 5 | 1 | 5 | 1 | 5 | 1 | 5 |
| GPT-5 | $94.0 \pm 3.50$ | $98.5 \pm 2.00$ | $85.5 \pm 5.00$ | $92.0 \pm 4.00$ | $79.5 \pm 5.50$ | $91.0 \pm 4.00$ | $74.0 \pm 6.00$ | $87.0 \pm 5.00$ |
| Claude Sonnet 4.5 | $91.0 \pm 4.50$ | $97.5 \pm 2.50$ | $76.5 \pm 6.00$ | $83.0 \pm 5.00$ | $75.5 \pm 6.00$ | $88.5 \pm 4.50$ | $66.0 \pm 6.50$ | $80.5 \pm 5.51$ |
| Gemini 2.5 Flash | $83.0 \pm 6.00$ | $91.0 \pm 4.50$ | $74.5 \pm 6.50$ | $83.0 \pm 5.50$ | $50.0 \pm 8.00$ | $74.5 \pm 6.00$ | $44.0 \pm 7.50$ | $66.0 \pm 6.50$ |
| Grok 4 Fast | $76.5 \pm 6.50$ | $94.0 \pm 3.50$ | $71.5 \pm 6.50$ | $89.5 \pm 4.50$ | $48.5 \pm 7.00$ | $77.0 \pm 6.00$ | $41.5 \pm 6.50$ | $72.0 \pm 6.01$ |
| Qwen3-Max-Preview | $83.5 \pm 5.50$ | $96.0 \pm 3.00$ | $75.0 \pm 6.01$ | $87.0 \pm 5.00$ | $62.0 \pm 6.50$ | $85.5 \pm 5.00$ | $56.5 \pm 6.50$ | $80.5 \pm 5.50$ |
| Deepseek-V3.2-Exp | $77.5 \pm 5.51$ | $91.0 \pm 4.00$ | $67.0 \pm 6.50$ | $79.5 \pm 6.00$ | $48.5 \pm 7.50$ | $73.0 \pm 6.50$ | $39.0 \pm 7.00$ | $61.5 \pm 7.00$ |
| gpt-oss-120b | $78.5 \pm 5.51$ | $91.0 \pm 4.00$ | $72.0 \pm 6.50$ | $83.0 \pm 5.50$ | $48.0 \pm 7.01$ | $68.0 \pm 6.51$ | $42.0 \pm 7.00$ | $60.5 \pm 7.50$ |
| Llama 4 Maverick | $55.0 \pm 7.00$ | $76.0 \pm 6.00$ | $47.0 \pm 6.50$ | $68.5 \pm 6.50$ | $46.5 \pm 7.00$ | $70.0 \pm 6.50$ | $37.5 \pm 7.00$ | $59.0 \pm 7.00$ |
| Qwen3-235B-A22B-Thinking-2507 | $74.0 \pm 6.50$ | $89.5 \pm 4.50$ | $66.0 \pm 6.51$ | $82.0 \pm 5.50$ | $36.5 \pm 7.00$ | $61.0 \pm 7.00$ | $33.0 \pm 6.50$ | $56.5 \pm 7.50$ |

## L.2. Example confidence intervals for Transform speedups

For the Transform track we report both static and dynamic speedups relative to `-O3`. Static speedups are obtained via `llvm-mca` and treated as deterministic. Dynamic speedups are estimated from repeated wall-clock measurements on fixed hardware.

For each kernel instance and mode (Direct or Copilot), we obtain $R$ repeated measurements of the `-O3` baseline and the LLM-optimized candidate (we use $R = 10$ in our experiments). From these runs we compute per-run dynamic speedups and report the sample mean $\bar{s}$ together with a 95% confidence interval obtained from the empirical variance across runs (assuming approximate normality for $R = 10$).

Table 18 illustrates representative 95% confidence intervals for GPT-5 on ten loop kernels from the T001 subset in Direct mode. Each entry reports the mean speedup and the 95% CI half-width. We omit the full table over all models and subsets for space; the pattern is similar elsewhere. Under our setup, run-to-run noise is small relative to the effect sizes of interest: for most kernels the CI half-width is only a few percent of the mean, and even in the most extreme case (K5) the interval is narrow enough to clearly distinguish a substantial speedup from noise.

## M. Qualitative case studies

### M.1. A Transform instance where GPT-5 outperforms `-O3`

We illustrate one representative Transform instance where the GPT-5 variant is *formally equivalent* to the `-O3` baseline yet consistently faster at run time.

**Setup and verification.** The instance comes from the "Ops" group in the Transform track. For each of raw, `-O3` ("golden"), and LLM-produced ("variant") IR, our toolchain compiles to a binary using the shared harness:

```
clang -no-pie raw.o    bench_utils.o -o raw.bin
clang -no-pie golden.o bench_utils.o -o golden.bin
clang -no-pie variant.o bench_utils.o -o variant.bin
```

*Table 16.* 95% bootstrap CI half-widths (in absolute probability) for Valid@{1,5}, Equiv@{1,5} on the Refactor-Normal track ($B = 1000$ bootstrap resamples). Each entry $w$ corresponds to a CI of the form $[p - w,\ p + w]$ for the point estimate $p$ in the corresponding cell of Table 4.

| Model | Valid@ | | Equiv@ | |
| --- | --- | --- | --- | --- |
| | 1 | 5 | 1 | 5 |
| GPT-5 | $42.0 \pm 7.00$ | $57.0 \pm 7.00$ | $22.5 \pm 5.50$ | $29.5 \pm 6.50$ |
| Claude Sonnet 4.5 | $48.0 \pm 7.00$ | $65.0 \pm 7.00$ | $24.0 \pm 6.00$ | $31.0 \pm 6.50$ |
| Gemini 2.5 Flash | $36.5 \pm 6.50$ | $62.5 \pm 7.00$ | $15.5 \pm 5.00$ | $28.5 \pm 6.50$ |
| Grok 4 Fast | $33.5 \pm 6.00$ | $56.5 \pm 6.50$ | $16.5 \pm 5.00$ | $27.5 \pm 6.50$ |
| Qwen3-Max-Preview | $24.0 \pm 5.50$ | $37.0 \pm 6.50$ | $10.5 \pm 4.00$ | $18.0 \pm 5.50$ |
| Deepseek-V3.2-Exp | $27.5 \pm 6.50$ | $57.0 \pm 7.00$ | $11.5 \pm 4.50$ | $24.0 \pm 6.00$ |
| gpt-oss-120b | $34.5 \pm 6.50$ | $59.0 \pm 6.51$ | $15.5 \pm 5.01$ | $24.5 \pm 6.00$ |
| Llama 4 Maverick | $35.5 \pm 6.50$ | $71.0 \pm 6.00$ | $16.0 \pm 5.50$ | $27.5 \pm 6.50$ |
| Qwen3-235B-A22B-Thinking-2507 | $22.0 \pm 6.67$ | $47.5 \pm 7.22$ | $8.5 \pm 4.44$ | $16.5 \pm 5.56$ |

*Table 17.* 95% bootstrap CI half-widths (in absolute probability) for Valid@{1,5} and Equiv@{1,5} on the Transform track ($B = 1000$ bootstrap resamples). Each entry $w$ corresponds to a CI of the form $[p - w,\ p + w]$ for the point estimate $p$ in the corresponding cell of Table 4.

| Model | Valid@ | | Equiv@ | |
| --- | --- | --- | --- | --- |
| | 1 | 5 | 1 | 5 |
| GPT-5 | $80.6 \pm 4.34$ | $97.0 \pm 2.00$ | $73.3 \pm 5.00$ | $92.7 \pm 3.00$ |
| Claude Sonnet 4.5 | $38.3 \pm 5.34$ | $65.0 \pm 5.33$ | $28.0 \pm 5.00$ | $50.0 \pm 5.34$ |
| Gemini 2.5 Flash | $71.6 \pm 5.48$ | $93.8 \pm 2.74$ | $31.5 \pm 5.49$ | $46.6 \pm 5.82$ |
| Grok 4 Fast | $41.7 \pm 5.34$ | $85.7 \pm 4.00$ | $31.7 \pm 5.33$ | $76.3 \pm 5.00$ |
| Qwen3-Max-Preview | $56.3 \pm 6.00$ | $90.6 \pm 3.34$ | $37.3 \pm 5.67$ | $75.3 \pm 5.00$ |
| Deepseek-V3.2-Exp | $56.3 \pm 5.67$ | $91.3 \pm 3.33$ | $15.3 \pm 4.33$ | $40.7 \pm 5.33$ |
| gpt-oss-120b | $60.9 \pm 5.45$ | $89.5 \pm 3.40$ | $28.2 \pm 5.11$ | $64.7 \pm 5.78$ |
| Llama 4 Maverick | $42.0 \pm 5.33$ | $69.3 \pm 5.33$ | $12.6 \pm 3.67$ | $27.0 \pm 5.33$ |
| Qwen3-235B-A22B-Thinking-2507 | $52.3 \pm 6.00$ | $82.0 \pm 4.33$ | $0.25 \pm 4.67$ | $0.51 \pm 5.67$ |

All three binaries pass the LLVM verifier:

```
opt -passes=verify v.ll -disable-output
```

and the -O3 baseline and GPT-5 variant are proved equivalent by Alive2:

```
alive-tv alive_input.ll -src-fn=src -tgt-fn=tgt --quiet
Transformation seems to be correct!
```

Finally, we run the harness at multiple settings. The first argument selects a data configuration (changing the array contents and therefore the checksum), while the second argument controls the outer loop count (changing the number of iterations but leaving the checksum unchanged for fixed data). All runs agree on the checksum, confirming semantic equivalence.

**IR before and after.** The kernel implements a clamping operator on a 1D array, conceptually equivalent to:

$$out[i] = clamp(in[i], low, high).$$

The GPT-5 variant uses a straightforward scalar loop:

```
define dso_local void @kernel_run(i32 noundef %0,
                                  ptr noundef %1, ptr noundef %2,
                                  double noundef %3, double noundef %4) #0 {
entry:
  br label %loop

loop:                                    ; preds = %latch, %entry
  %i     = phi i32 [ 0, %entry ], [ %i.next, %latch ]
  %cond  = icmp slt i32 %i, %0
  br i1 %cond, label %body, label %exit

body:                                    ; preds = %loop
```

*Table 18.* Example 95% confidence intervals for GPT-5 dynamic speedup over -O3 on ten loop kernels from the T001 Transform subset (Direct mode, $R = 10$ runs per binary). Each entry is reported as mean speedup $\pm$ 95% CI half-width.

| Kernel (GPT-5, T001 Loops, Direct) | Speedup (95% CI) |
| --- | --- |
| T001_Loops_001 | $4.06 \pm 0.86$ |
| T001_Loops_002 | $1.01 \pm 0.27$ |
| T001_Loops_003 | $0.76 \pm 0.13$ |
| T001_Loops_004 | $0.18 \pm 0.04$ |
| T001_Loops_005 | $2.92 \pm 1.35$ |
| T001_Loops_006 | $0.74 \pm 0.19$ |
| T001_Loops_007 | $0.55 \pm 0.26$ |
| T001_Loops_008 | $1.74 \pm 0.45$ |
| T001_Loops_009 | $0.89 \pm 0.20$ |
| T001_Loops_010 | $0.73 \pm 0.28$ |

```llvm
  %idx     = sext i32 %i to i64
  %in.ptr  = getelementptr inbounds double, ptr %1, i64 %idx
  %val     = load double, ptr %in.ptr, align 8
  %lt      = fcmp olt double %val, %3
  %gt      = fcmp ogt double %val, %4
  %sel_hi  = select i1 %gt, double %4, %val
  %out     = select i1 %lt, double %3, double %sel_hi
  %out.ptr = getelementptr inbounds double, ptr %2, i64 %idx
  store double %out, ptr %out.ptr, align 8
  br label %latch

latch:                                            ; preds = %body
  %i.next = add nsw i32 %i, 1
  br label %loop, !llvm.loop !13

exit:                                             ; preds = %loop
  ret void
}
```

The -O3 baseline, by contrast, applies several loop transformations (including partial unrolling and an epilogue), resulting in multiple loop headers, epilogue blocks, and additional branches:

```llvm
define dso_local void @kernel_run(i32 noundef %0,
                                  ptr nocapture noundef readonly %1,
                                  ptr nocapture noundef writeonly %2,
                                  double noundef %3, double noundef %4)
                                  local_unnamed_addr #0 {
  %6 = icmp sgt i32 %0, 0
  br i1 %6, label %.lr.ph.preheader, label %._crit_edge

.lr.ph.preheader:
  %wide.trip.count = zext nneg i32 %0 to i64
  %xtraiter = and i64 %wide.trip.count, 1
  %7 = icmp eq i32 %0, 1
  br i1 %7, label %._crit_edge.loopexit.unr-lcssa,
            label %.lr.ph.preheader.new

.lr.ph.preheader.new:
  %unroll_iter = and i64 %wide.trip.count, 2147483646
  br label %.lr.ph

; ... two-way unrolled main loop and scalar epilogue ...
; (see artifact for the full function)

._crit_edge:
  ret void
}
```

Alive2 confirms that both implementations are equivalent for all inputs, but they present very different trade-offs between loop structure and code size.

**Timing results.** We measure run time using the shared harness and report the average of three runs for each configuration. Table 19 summarizes the results for three representative settings; in each case the checksums of golden and variant match exactly.

*Table 19.* Representative timings for the Transform case study. The first argument selects the data configuration (which changes the checksum); the second argument controls the outer loop count. Times are averages over three runs.

| Args | Checksum | Loops | Golden (ms) | Variant (ms) | Speedup |
|------|----------|-------|-------------|--------------|---------|
| (3, 2) | 0x9098… | $10^4$ | 0.364 | 0.247 | $1.47\times$ |
| (3, 3) | 0x9098… | $10^6$ | 32.80 | 21.81 | $1.50\times$ |
| (2, 3) | 0x0408… | $10^6$ | 8.31 | 5.49 | $1.52\times$ |

Across both data configurations and loop counts, the GPT-5 variant is consistently about $1.5\times$ faster than the -O3 baseline while preserving semantics under verifier, Alive2, and checksum guards.

**Why is the LLM variant faster?** The speedup in this case can be explained by two complementary effects.

- **Simpler loop structure.** The GPT-5 variant uses a single, non-unrolled loop with one induction variable and a compact body. The -O3 version performs partial unrolling with an epilogue, introducing extra control-flow edges, PHI nodes, and scalar cleanup code. On this microkernel and target CPU, the additional branches and bookkeeping do not amortize well, so the smaller loop executes more efficiently.

- **More regular control flow for clamping.** Both versions implement the same clamp logic, but the variant expresses it using two chains of select instructions guarded by simple comparisons. The baseline uses additional conditional branches and PHI merges inside each unrolled iteration. The variant therefore reduces branch misprediction risk and keeps the hot path short, which is reflected in the roughly $1.5\times$ reduction in measured run time.

This example illustrates the kind of "superoptimizing" opportunity that CIRBench exposes: even under strict semantic constraints, LLMs can sometimes discover IR rewrites that trade a more complicated, heavily transformed loop for a simpler implementation that runs faster on real hardware. At the same time, such wins are rare in aggregate (cf. subsection 4.5), highlighting the need for mechanisms that can predict when a proposed rewrite is likely to behave like this case rather than a neutral or negative one.

# N. Walkthrough Example

To make the four tracks in CIRBench more concrete, we present a single walkthrough example that reuses the same LLVM IR fragment across Analysis, Refactor, Repair, and Transform. This example is illustrative and not part of the evaluation set, but it matches the style and difficulty of typical CIRBench instances.

## N.1. Shared IR fragment

Consider the following kernel, which sums the first n elements of an array and adds a scalar bias:

```
define i32 @sum_bias(i32* %arr, i32* %bias, i32 %n) {
entry:
  br label %loop

loop:
  %i    = phi i32 [0, %entry], [%i.next, %body]
  %sum  = phi i32 [0, %entry], [%sum.next, %body]
  %cmp  = icmp slt i32 %i, %n
  br i1 %cmp, label %body, label %exit
```

```
body:
  %idx    = sext i32 %i to i64
  %ptr    = getelementptr inbounds i32, i32* %arr, i64 %idx
  %val    = load i32, i32* %ptr
  %bval   = load i32, i32* %bias
  %tmp    = add nsw i32 %val, %bval
  %sum.next = add nsw i32 %sum, %tmp
  %i.next  = add nuw i32 %i, 1
  br label %loop

exit:
  ret i32 %sum
}
```

This function has a single natural loop with header `loop`, a loop body `body`, an exit block `exit`, and a loop-invariant scalar `%bias` that is redundantly reloaded on each iteration. We next sketch how the four tracks in CIRBench would formulate tasks on this IR.

### N.2. Analysis: control and data-flow queries

In the Analysis track, the model is asked to recover properties that a compiler would compute before optimization. For the example above, the following questions are representative:

- **CFG.** "List all basic blocks and the directed edges between them." The ground-truth JSON encodes blocks {entry, `loop`, `body`, `exit`} and edges entry→loop, loop→body, loop→exit, body→loop.

- **Loop structure.** "What is the loop header, latch, and exit block?" The oracle answer marks `loop` as header and latch, and `exit` as the unique exit.

- **Alias.** "Do `%arr` and `%bias` ever alias?" Since they are independent arguments, the correct label is `NoAlias`.

- **Basic properties.** "How many `load` and `store` instructions does the function contain?" Here the answer is four loads and zero stores.

Model outputs are normalized to canonical JSON and compared by EM and F1, as described in subsection 3.2.

### N.3. Refactor: applying a single pass

In the Refactor-Normal track, the same IR may appear with the instruction

> "Apply LICM (loop-invariant code motion) to this function."

The reference IR is obtained by running the corresponding LLVM pass. For this example, LICM hoists the invariant load from `%bias` out of the loop into a preheader:

```
entry:
  %bval.pre = load i32, i32* %bias
  br label %loop

loop:
  %i      = phi i32 [0, %entry], [%i.next, %body]
  %sum    = phi i32 [0, %entry], [%sum.next, %body]
  ...
body:
  ...
  ; use %bval.pre instead of reloading %bias
  %tmp    = add nsw i32 %val, %bval.pre
  ...
```

The model is free to return any IR that is verifier-clean and functionally equivalent; Equiv@k scores semantic success, while EM@k indicates whether the model exactly matches the reference rewrite after normalization.

In Refactor-Reverse, the before/after pair corresponding to this change is given and the model is asked to identify that LICM was applied.

### N.4. Repair: fixing injected errors

For the Repair track, we introduce structured corruption into the same function. For instance, we may break the PHI node for `%sum`:

```
loop:
  %i   = phi i32 [0, %entry], [%i.next, %body]
  %sum = phi i32 [0, %entry]                ; missing incoming from %body
  ...
```

LLVM's verifier reports an error such as *"PHI node has wrong number of incoming values"*. In *Hint* mode, this diagnostic and the block name are included in the prompt; in *Hard* mode, the model only sees the raw IR.

A successful repair must (i) restore verifier cleanliness and (ii) preserve the original semantics, e.g. by adding the missing incoming edge `[%sum.next, %body]`. Equiv@k distinguishes semantically correct repairs from candidates that only satisfy the verifier.

### N.5. Transform: optimizing for performance

Finally, the same kernel can be used as a Transform instance. The prompt states the target platform and asks the model to propose semantics-preserving IR-to-IR rewrites that improve runtime. For this example, profitable rewrites include:

- reusing the hoisted `%bval.pre` value as in the LICM refactoring;

- unrolling the loop by a small factor (e.g., 2) to reduce branch overhead;

- combining scalar loads into a short vector loop when the hardware supports it.

The resulting IR is passed through the layered evaluation pipeline: the verifier filters out malformed candidates, Alive2 and the harness check semantic equivalence, and static (`llvm-mca`) and dynamic measurements compare performance against `-O3`.

This walkthrough illustrates how a single IR fragment can support multiple, complementary questions in CIRBench: from recovering compiler analyses, through controlled pass-level refactorings and structured repair, to open-ended performance-oriented rewriting, all at the same IR boundary where production compilers make optimization decisions.

## O. Further Analysis

This section collects several exploratory observations and hypotheses suggested by CIRBench. They are not central claims of the paper, but we hope they can help shape future work on IR-centric LLMCompiler systems.

**Smaller models for IR analysis.**   On the Analysis track, the gap between the strongest and mid-tier models is modest compared to generative tracks, while the cost difference is substantial. This suggests that IR analysis may be an attractive target for smaller, specialized models: extracting CFGs, dominance, or simple properties from IR is structurally regular, and could plausibly be handled by fine-tuned graph or sequence encoders that run much faster than large general-purpose LLMs. Our current results are consistent with this view, but a systematic comparison between small dedicated analyzers and large LLMs remains open.

**Open questions on Analysis–Transform coupling.**   Intuitively, weaknesses in IR analysis should limit the safety and quality of downstream Transform suggestions. While our initial correlation checks are inconclusive. Causally disentangling "better analysis" from "stronger generation" will likely require controlled studies where Transform is given either oracle or model-predicted analysis facts.

**Sampling gaps and IR coverage.** Across Repair, Refactor-Normal, and Transform, Equiv@5 is consistently higher than Equiv@1. This indicates that models do benefit from sampling diversity: different candidates often touch different locations or pursue different rewrite strategies. One possible interpretation is that pretraining has not exposed models to enough LLVM IR to make single-sample behavior robust; another is that current decoding policies are not well matched to IR editing. Our results do not distinguish these explanations, but they highlight sampling as an important degree of freedom for IR-centric deployments.

**Model scale and IR optimization ability.** Transform performance does not appear to grow monotonically with parameter count. Some mid-sized models approach or even surpass larger ones on certain tracks, while lagging on others. This is unsurprising given differences in training data, system prompts, and post-training objectives, but it suggests that "bigger is better" is not a reliable heuristic for IR-level optimization. A more principled study of how data, architecture, and alignment influence IR reasoning ability is an important direction.

**Breadth and depth of optimizations.** Manual inspection of Transform outputs reveals that many successful rewrites rely on relatively standard scalar optimizations (e.g., constant folding, algebraic simplification, strength reduction) rather than more aggressive techniques such as vectorization or complex loop transformations. Some models achieve relatively high Transform correctness but mainly perform low-level cleanups; others occasionally attempt deeper restructurings but incur more semantic errors. Quantifying this "optimization repertoire" more precisely—for example, by matching rewrites to known pass patterns—remains future work.

**Direct vs. Copilot behavior.** Our Direct vs. Copilot comparison shows a characteristic pattern: passing model-produced IR through `-O3` tends to truncate the worst slowdowns and move medians toward the compiler baseline, but also moderates some of the largest speedups. In other words, Copilot behaves like a risk-averse controller that trades away part of the upside in exchange for fewer catastrophic regressions. Understanding how to design such controllers—possibly with learned policies that decide when to trust the model, when to fall back, and when to combine suggestions—seems crucial for making LLMCompiler systems robust in practice.

**Pass-level literacy, attributes, and metadata.** The low EM@k scores on Refactor-Normal, despite reasonable Equiv@k, suggest that current models often ignore or mishandle information outside the function body: attributes, calling conventions, global variables, and metadata. These IR-wide invariants matter both for optimization correctness and for fidelity to named passes. A promising avenue is to give models more structured access to such information (e.g., separate views for attributes and metadata) or to train explicitly on tasks that require preserving them.

**Module-level IR and cross-function constraints.** Most current LLM-for-compilers work, including CIRBench's main experiments, operates at the function level. Our small module-level subset already exposes additional challenges: function summaries, cross-call invariants, and linkage semantics become relevant, and simple local reasoning is no longer sufficient. We are not aware of systematic benchmarks focused on module-level IR optimization with LLMs, and view this as a largely unexplored but practically important setting.

**Scaling with IR size and structural errors.** On Transform, correctness degrades as functions become larger and control-flow more complex. Longer instances exhibit higher rates of dominance and CFG errors, as well as more frequent mistakes in PHI nodes and induction-variable updates. This supports the view that "keeping the skeleton" of control flow is within reach, but that detailed invariants about dominance and data flow increasingly slip as IR grows. Designing representations and prompts that scale more gracefully with function size is an open challenge.

**IR version drift.** LLVM IR is not a single fixed language: instructions, attributes, and verifier rules evolve across versions. CIRBench fixes a particular LLVM version and toolchain, but real-world deployments may encounter mixtures of IR from different vintages. Our benchmark does not attempt to model this drift, and we caution against overgeneralizing results across LLVM versions. Explicitly conditioning models on IR dialect or version, or training them to handle multiple dialects, may be necessary for long-term robustness.

**Hard-core instances and headroom.** Across tracks, a small but non-trivial subset of instances remains unsolved by all nine models even at $k = 5$. These "hard-core" cases tend to concentrate on long IR with complex control flow or multiple

interacting errors. We do not attempt to fully characterize them here, but they provide natural stress tests and potential targets for future method development and curriculum design.

**Static vs. dynamic performance signals.**    Where both static (`llvm-mca`) and dynamic measurements are available, we observe a noisy but generally positive correlation between static and dynamic speedups. This suggests that static cost models could act as inexpensive filters in LLMCompiler pipelines, identifying promising candidates before committing to more expensive runtime evaluation. A more detailed analysis of when static estimates are reliable, and how they fail, is an interesting direction.

**Complementarity across model families.**    Per-instance results reveal that different model families succeed on partially disjoint subsets of CIRBench. A hypothetical oracle that, for each instance, selects the best candidate among all nine models substantially outperforms any individual model. This indicates that current models capture complementary IR patterns and failure modes, and that ensemble or model-selection approaches could yield non-trivial gains over single-model baselines.

**Task- and subset-specific behavior.**    Within the Transform track, different subsets exhibit distinct difficulty profiles. Loop-centric and ML/DL kernels are generally more amenable to successful rewrites, while algorithmic and module-level kernels remain harder, and the Challenge subset concentrates the most difficult, high-headroom patterns. These differences hint at interactions between pretraining distributions, IR structure, and optimization difficulty, and suggest that future work may benefit from task-specific modeling choices.

**Verifier hints and error types.**    On the Repair track, we find that verifier hints do not benefit all error classes equally. Certain structural errors—such as broken PHI nodes or type mismatches—gain substantially from explicit messages and locations, whereas more semantic issues involving UB or subtle attribute interactions remain difficult in both Hint and Hard modes. This points toward a more nuanced picture of "diagnostics as supervision" and suggests that future systems may need richer error feedback or learned diagnostic models.

## P. Benchmark Sources and Licenses

CIRBench is constructed from a small number of well–established compiler and program benchmarks. Table 20 summarizes the main assets we use, the tracks in which they appear, and their original licenses. In all cases, we only use compiled LLVM IR fragments or small kernels derived from the original suites; we do not redistribute full projects, binaries, or datasets.

*Table 20.* Upstream assets used in CIRBench and their licenses.

| Asset | Tracks used | License |
|---|---|---|
| IR-Optset | Analysis, Repair, Refactor | MIT |
| LFK | Transform | BSD 2-Clause |
| LLVM Test Suite | Analysis | Apache 2.0 |
| PolyBench | Transform, Repair | MIT |
| TSVC2 | Transform | Illinois Open Source License |
| BiSheng Compiler | Transform | Apache 2.0 |
| LLVM PGO tooling | Transform | Apache 2.0 |
| CodeNet | Transform | Apache 2.0 |

**CIRBench license.**    The CIRBench benchmark itself (including our generated IR instances, toolchain scripts, and evaluation code) is released under the Apache 2.0 license. This license applies to our contributions and does not alter the original licenses of the upstream assets in Table 20. Researchers building on CIRBench should respect both the Apache 2.0 terms for CIRBench and the licenses of the underlying benchmarks.

## Q. Additional Hardware Back-ends

To check that our Transform results are not specific to the AMD Zen 4 machine used in the main text, we repeat the evaluation on a desktop Intel CPU. All static `llvm-mca` estimates and dynamic speedups in Table 21 are measured on an 11th Gen Intel® Core™ i9-11900K @ 3.50 GHz system (x86_64 ISA, 8 physical cores / 16 hardware threads, 16 MiB shared L3 cache), with pinned cores and the same compilation flags and timing protocol as in the main AMD experiment.

*Table 21.* Performance of nine models on the Transform task under direct and copilot modes.

| Model | direct | | | | | | copilot | | | | | |
|---|---|---|---|---|---|---|---|---|---|---|---|---|
| | mca | | | perf | | | mca | | | perf | | |
| | min | med | max | min | med | max | min | med | max | min | med | max |
| **GPT-5** | 0.12 | 0.83 | 1.82 | 0.09 | 0.69 | 2.92 | 0.10 | 0.87 | 2.01 | 0.32 | 0.92 | 3.10 |
| **Claude Sonnet 4.5** | 0.23 | 0.80 | 1.22 | 0.22 | 0.83 | 5.68 | 0.15 | 0.78 | 1.08 | 0.33 | 0.90 | 4.13 |
| **Gemini 2.5 Flash** | 0.19 | 0.77 | 1.22 | 0.22 | 0.81 | 2.94 | 0.26 | 0.78 | 1.23 | 0.41 | 0.90 | 2.63 |
| **Grok 4 Fast** | 0.12 | 0.83 | 1.64 | 0.08 | 0.54 | 4.19 | 0.17 | 0.88 | 2.10 | 0.38 | 0.95 | 4.75 |
| **Qwen3-Max-Preview** | 0.23 | 0.83 | 2.05 | 0.11 | 0.62 | 4.45 | 0.26 | 0.85 | 1.72 | 0.35 | 0.94 | 4.25 |
| **Deepseek-V3.2-Exp** | 0.19 | 0.79 | 1.20 | 0.12 | 0.65 | 7.72 | 0.26 | 0.81 | 1.18 | 0.39 | 0.94 | 4.29 |
| **gpt-oss-120b** | 0.12 | 0.82 | 1.68 | 0.12 | 0.56 | 4.25 | 0.17 | 0.84 | 1.72 | 0.20 | 0.96 | 4.37 |
| **Llama 4 Maverick** | 0.26 | 0.82 | 1.17 | 0.17 | 0.41 | 8.17 | 0.22 | 0.79 | 1.02 | 0.40 | 0.96 | 5.00 |
| **Qwen3-235B-A22B-Thinking-2507** | 0.12 | 0.78 | 1.21 | 0.13 | 0.61 | 4.71 | 0.27 | 0.81 | 1.06 | 0.38 | 0.91 | 4.13 |

Across the nine models, the pattern on this Intel backend closely matches the Zen 4 results: median speedups remain below $1.0\times$ in both *direct* and *copilot* modes, while copilot consistently shrinks the most negative tail and slightly improves the upper tail. These cross-platform results support our main conclusion that current LLM-based IR rewrites are fragile optimizers—capable of occasional large gains but also frequent slowdowns—and that using them in a "copilot" configuration is generally safer than applying them directly.

