# OpenReview forum: "CIRBench: Evaluating Large Language Models as LLVM IR Optimizers"
_ICML.cc/2026/Conference — ICML 2026 spotlight_

### Official Review · Reviewer_exAS · 2026-03-11

**Soundness:** 3
**Presentation:** 3
**Significance:** 3
**Originality:** 3
**Overall Recommendation:** 5
**Confidence:** 4

**Summary:**

This paper introduces CIRBench, a benchmark for evaluating LLMs on LLVM intermediate representation tasks. It is organized into four tracks: Analysis (extracting structural/semantic properties), Repair (fixing corrupted IR), Refactor (reproducing optimization passes), and Transform (producing performance-improving rewrites). The benchmark draws IR from established compiler test suites and evaluates nine frontier LLMs using layered correctness metrics (Valid@k, Equiv@k). Key findings include: current LLMs achieve moderate IR understanding (Analysis EM 41-70%) but struggle with generation; a large gap persists between producing verifier-clean IR and semantically correct IR; and on Transform, median speedups remain below 1.0x relative to -O3. The benchmark and toolchain are open-sourced.

**Compliance With Llm Reviewing Policy:**

Affirmed.

**Final Justification:**

The response has addressed the questions I raised. I believe my previous score is appropriate and will maintain my original rating. Thank you.

**Key Questions For Authors:**

1. Have you considered evaluating against weaker optimization baselines (-O1, -O2) for the Transform track? This would better contextualize the findings. If results show LLMs can contribute at lower optimization tiers, I would raise my score.
2. How do you handle cases in Refactor-Reverse where multiple passes could produce similar transformations? Is there analysis of potential false negatives from strict exact-match evaluation?
3. Would including a fine-tuned or IR-specialized baseline help characterize the headroom and determine whether observed limitations are fundamental or reflect lack of domain-specific training?

**Limitations:**

The authors discuss several limitations including contamination concerns, function-level scope, and LLVM version drift. Additional considerations include sensitivity to prompt design (model rankings may change with model-specific prompts) and the gap between static cost models and actual dynamic performance (noted as "noisy" in Appendix O). Overall the limitations discussion is reasonable.

**Strengths And Weaknesses:**

Strengths:
- The evaluation methodology is rigorous with well-layered metrics. The Valid@k vs. Equiv@k distinction (Section 3.2) is particularly valuable, separating syntactic well-formedness from semantic correctness. The use of Alive2 for formal equivalence checking and bootstrap confidence intervals (Appendix L) strengthen the claims.
- CIRBench fills a genuine gap with its unified, multi-track evaluation framework spanning analysis through optimization. The four-track progressive design and the copilot vs. direct comparison (Section 4.5) are well-motivated and novel.
- The finding that IR understanding does not imply IR editing ability (Section 4.3) is an important insight for the community. The open-source toolchain substantially increases potential impact.
- The paper is well-organized with a clear four-track structure (Figure 1). The corruption strategies for Repair (Table 1) are thoughtfully designed with controlled difficulty levels.

Weaknesses:
- Contamination concerns are significant and only partially addressed (Appendix I). The benchmark sources have been publicly available for years, and models may have memorized optimization patterns. A contamination audit would strengthen confidence in result generalizability.
- The Transform track compares against -O3 only. Comparing against weaker baselines (-O1, -O2) would better characterize where LLMs can contribute in the optimization spectrum.
- Some critical details (exact prompts, normalization pipeline, analysis question types) are relegated to appendices that are essential for understanding the evaluation. Better integration in the main text would improve accessibility.

---

> ### Author Rebuttal · Authors · 2026-03-28
>
> Thank you for the constructive suggestions and positive assessment of the benchmark design, layered metrics, and open-source toolkit. We address your points below.
>
> 1. **Contamination / provenance.**
>
> We agree this should be stated more clearly. CIRBench is not claimed to be contamination-free: it is built from long-standing public suites, and we do not interpret it as a decontaminated reasoning benchmark. Our claim is narrower: CIRBench evaluates reliability at the LLVM IR boundary under verifier / equivalence / performance constraints. Instances are not evaluated as raw source programs; they are compiled to LLVM IR and then undergo slicing, normalization, task-specific construction, deduplication, UB filtering, and timing filtering. We view this as reducing trivial source-level transfer, while not ruling out overlap. In the revision, we will move this disclaimer into the main text and add a compact provenance table.
>
> **Table A. Provenance / contamination transparency**
>
> | Source family                           | Representative source                | Track(s) used in                      | # instances | Approx. public vintage |
> | --------------------------------------- | ------------------------------------ | ------------------------------------- | ----------- | ---------------------- |
> | LLVM-native / IR-oriented public suites | IR-Optset; LLVM test suite           | Analysis, Repair, Refactor, Transform | 500         | 2002–2025              |
> | Classical benchmark suites              | LFK; TSVC; PolyBench; CodeNet        | Transform                             | 120         | 1986–2021              |
> | Curated / hand-written / mined kernels  | hand-written kernels; T005 challenge | Transform                             | 180         | -                      |
>
> 2. **`-O1/-O2` baselines.**
>
> We agree and will add same-harness comparisons against `-O1/-O2/-O3`. Our current main conclusion is relative to `-O3`, but the broader pattern is similar under weaker baselines: models can occasionally achieve large wins, yet robust median gains remain difficult. We report speedup as runtime(`-Ok`) / runtime(candidate), so values above 1 indicate improvement over the compiler baseline.
>
> **Table B. Transform candidate performance relative to different compiler baselines**
>
> | Setting       | vs. `-O1` min | vs. `-O1` med | vs. `-O1` max | vs. `-O2` min | vs. `-O2` med | vs. `-O2` max | vs. `-O3` min | vs. `-O3` med | vs. `-O3` max |
> | ------------- | ------------- | ------------- | ------------- | ------------- | ------------- | ------------- | ------------- | ------------- | ------------- |
> | GPT-5 Direct  | 0.08          | 0.88          | 4.89          | 0.08          | 0.87          | 4.89          | 0.08          | 0.85          | 4.89          |
> | GPT-5 Copilot | 0.19          | 1.00          | 4.47          | 0.20          | 0.99          | 3.23          | 0.19          | 0.99          | 3.23          |
>
> 3. **Refactor-Reverse exact match.**
>
> Exact match is intentional because this is a closed-set pass-identification task over 20 candidate LLVM passes. Each instance is extracted from a single-pass `-print-changed` event, filtered to remove trivial edits, and validated by equivalence checking. We agree, however, that exact pass identity is stricter than semantic plausibility and that some passes can yield overlapping visible effects (e.g., GVN vs. InstSimplify). We will therefore clarify the task definition in the main text and add a short confusion analysis, so this metric is read as pass identification rather than as a complete measure of rewrite correctness.
>
> 4. **IR-specialized baselines.**
>
> The current submission already includes three additional open-weight baselines in Appendix J under the same prompting / decoding protocol. To address your point more directly within the rebuttal window, we will also add a preliminary Transform-only result for LLMCompiler-7B under the same CIRBench harness, together with practical failure-mode breakdowns (e.g., context-length overflow and invalid generations), as an initial IR-oriented reference point.
>
> **Table C. Preliminary IR-oriented baseline on Transform**
>
> | Model          | Valid@1 | Equiv@1 | Direct perf min | Direct perf med | Direct perf max |
> | -------------- | ------- | ------- | --------------- | --------------- | --------------- |
> | LLMCompiler-7B | 34.3%   | 2.7%    | 0.361           | 0.937           | 1.03            |
>
> 5. **Main-paper clarity.**
>
> We agree that several appendix-only details should be moved forward. In the revision, we will promote the contamination disclaimer, the added `-O1/-O2` context, the Refactor-Reverse task definition, and the note on additional baselines into the main paper.
>
> These additions improve clarity, but do not alter the main conclusion: CIRBench remains a correctness-aware LLVM IR benchmark, and current LLMs are still substantially stronger at IR understanding than at reliable semantics-preserving IR rewriting.

---

> > ### Author Rebuttal · Reviewer_exAS · 2026-04-03
> >
> > The additional experiment provided by the user resolved the vast majority of my issues; therefore, I have decided to maintain my positive rating.

---

### Official Review · Reviewer_5hqD · 2026-03-13

**Soundness:** 2
**Presentation:** 3
**Significance:** 3
**Originality:** 3
**Overall Recommendation:** 4
**Confidence:** 3

**Summary:**

This paper introduces CIRBench, a benchmark for evaluating large language models on LLVM IR-level compiler tasks rather than source-level coding tasks. The benchmark contains 800 curated instances divided into four tracks: Analysis (predicting IR properties such as CFG, loops, aliasing, and dominators), Repair (fixing invalid IR), Refactor (applying a specified compiler pass), and Transform (open-ended performance-oriented IR rewrites). The central framing is that IR-level evaluation should impose compiler-grade semantic constraints, so the benchmark layers verification, equivalence checking, and performance measurement rather than relying on surface-form outputs alone.

The core methodological contribution is therefore not a new learning algorithm, but a benchmarking and evaluation protocol. The paper constructs instances from existing suites such as IR-Optset, the LLVM test suite, LFK, TSVC, and PolyBench, normalizes them, assigns oracle labels where appropriate, and evaluates generated IR with a layered pipeline using the LLVM verifier, Alive2 when applicable, checksum-based execution fallback in some cases, and static/dynamic performance measurement for the Transform track. The authors also stratify instances by difficulty and provide an open-source toolkit intended to support reproducible evaluation.

Empirically, the paper evaluates six mainstream LLMs across all four tracks under fixed prompt templates and standardized decoding settings. The headline result is that current models are still brittle on IR reasoning and rewriting: Analysis EM ranges roughly from 41% to 70%, Repair shows a substantial Valid–Equiv gap, Refactor remains very weak with near-zero exact pass execution accuracy, and Transform—despite some large best-case wins—shows median dynamic performance below the -O3 compiler baseline for every model in direct mode. GPT-5 is reported as the strongest overall model, especially on Transform equivalence, but even it does not produce median speedups above the compiler baseline.

The paper’s central contribution, as I read it, is a correctness-aware IR benchmark that exposes the difference between “plausible IR editing” and “reliable semantics-preserving compiler behavior.” That is a useful and timely problem setting. However, the paper is still primarily a benchmark curation and evaluation paper, and its acceptance case depends heavily on whether the protocol is sufficiently rigorous, whether the benchmark is clearly differentiated from prior IR-level evaluation efforts, and whether the empirical conclusions are supported with enough methodological transparency.

**Compliance With Llm Reviewing Policy:**

Affirmed.

**Key Questions For Authors:**

1. How often is equivalence established formally versus only through fallback testing, and how does that affect the main conclusions?
This is the most important reliability issue. The paper states that Alive2 is used when it returns a definite result, but otherwise the evaluation falls back to checksum-based harness testing, especially for module-level tasks or tool-limited cases. That means “Equiv” is not a single homogeneous guarantee. This is serious because the benchmark’s main claim is precisely about layered semantic correctness; if the strongest reported results rely heavily on weaker fallback mechanisms, the interpretation changes substantially. At minimum, the authors should provide a clear breakdown—by track, difficulty, and perhaps by model—of the fraction of instances evaluated under formal equivalence versus fallback testing, and ideally show that the main conclusions remain stable on the subset with the strongest semantic guarantees.

2. How robust are the benchmark results to contamination and memorization concerns? The authors openly state that they do not guarantee absence of pretraining contamination and that the benchmark is built from long-standing public suites. This is serious because, in a benchmark paper, the field needs confidence that the benchmark measures the intended capability rather than partly reflecting prior exposure. The issue is especially important for Analysis and possibly Refactor-like tasks, where recognizable structures may overlap with pretraining data. At minimum, the authors should quantify the risk better: perform stronger provenance analysis, add more recent or synthetic splits, analyze performance on more transformed or less canonical subsets, or isolate subsets with lower memorization risk and show whether the conclusions persist.

3. Why should ICML view CIRBench as clearly differentiated from prior IR-level evaluation efforts rather than an incremental aggregation benchmark? The paper cites prior work on IR-level understanding, IR optimization, and IR-focused datasets, and claims there is still no single benchmark spanning both analysis and semantics-preserving optimization with layered correctness. That may be true, but the paper needs to argue this distinction more sharply. This is serious because benchmark papers are judged heavily on whether they create a new standard, not just a more polished collection of existing task types. At minimum, the authors should produce a very explicit comparison table against the closest prior datasets/benchmarks, showing what is genuinely missing in existing evaluations and why the combination in CIRBench changes research behavior rather than merely broadening coverage.

**Limitations:**

The paper does acknowledge some limitations and includes a brief impact statement, but the discussion is not sufficient.

The most important missing piece is a more explicit treatment of evaluation validity. The paper should clearly discuss the consequences of using a mixed correctness regime, especially the gap between formal equivalence and checksum-based fallback testing, and explain what kinds of semantic errors may still slip through in the weaker regime. It should also discuss how much of the benchmark falls into each regime and how that constrains interpretation of cross-track comparisons.

The paper should also expand its discussion of contamination and benchmark provenance. Saying that contamination is an unresolved limitation is honest, but not enough for a benchmark paper. The authors should explain how this limitation may bias individual tracks differently, which claims remain robust despite possible overlap, and what future benchmark versions would need to do to improve trustworthiness.

Finally, the societal impact discussion is too generic. The paper mentions that better compilation could accelerate harmful workloads, but that is a shallow statement. A stronger discussion should address benchmark misuse for unsafe automated optimization, risks of overtrusting LLM-generated IR in systems contexts, the danger of equating verifier-passing outputs with semantic safety, and the possibility that this benchmark could encourage overly optimistic deployment claims if its caveats are ignored.

**Strengths And Weaknesses:**

The paper’s strongest aspect is that it tackles a real and important evaluation gap. Moving from source-level code generation to IR-level compiler manipulation is a meaningful shift, and the four-track design is more comprehensive than a narrow one-task benchmark. The layered correctness protocol is also directionally correct: using verification, equivalence checks, and performance together is much more appropriate for compiler-facing evaluation than relying on functional tests or textual similarity alone. The empirical takeaway is also useful and non-trivial: current LLMs can sometimes generate impressive isolated wins, but they remain unreliable and generally fail to beat the conventional compiler on median runtime, which is exactly the kind of sobering benchmark result the field needs. These aspects support the paper’s significance and make the benchmark potentially valuable to researchers working at the ML-for-compilers interface.

That said, the paper has several reliability weaknesses that materially affect its ICML case. The biggest one is that the semantic oracle is not uniform: some cases use Alive2, while others fall back to checksum-based harness testing when Alive2 is inconclusive or not applicable. That is understandable in practice, but the paper does not make sufficiently clear in the main text how often each regime is used, on which subsets, and how much the conclusions depend on this fallback. As a result, “Equiv” does not correspond to a single consistent guarantee across the benchmark. A second issue is contamination: the authors explicitly acknowledge that the benchmark is derived from long-standing public suites and does not guarantee absence of pretraining overlap, which weakens the strength of any claims about genuine IR reasoning rather than memorization or partial recall. A third issue is that several important results are harder to interpret than they should be. For example, Table 5 reports large overlapping error counts, but the counting unit and denominator are not clear enough in the main text, so it is difficult to judge what those numbers actually mean. These weaknesses do not invalidate the benchmark, but they do reduce confidence in the strength and cleanliness of the empirical conclusions.

The writing quality is mixed. On the positive side, the paper is generally well organized, the problem setup is easy to follow, and the four tracks are motivated in a coherent way. The high-level narrative—Analysis, Repair, Refactor, Transform—maps well to compiler responsibilities, and the main experimental findings are presented in a readable order. However, many of the most important methodological qualifications are pushed into appendices, while the main paper sometimes reads as though the protocol is cleaner than it actually is. For a benchmark paper, that is a problem. The reader needs immediate clarity on what is formally checked, what is only test-checked, how much of the benchmark is module-level versus function-level, how often fallback mechanisms are invoked, and how robust the performance claims are across hardware. The paper is therefore readable, but not yet as sharp and self-contained as an ICML-strength benchmark paper should be.

The work is respectable but not especially strong by top-tier standards. The novelty lies in the combination and formalization of tasks, the benchmark construction, and the layered evaluation protocol—not in a new model, learning principle, or theoretical insight. That can still be enough for acceptance if the benchmark is clearly indispensable and methodologically airtight. Here, however, the distinction from the closest prior IR-level datasets and evaluations is not yet articulated with enough precision in the main paper, and the benchmark’s most distinctive claims are partly weakened by contamination uncertainty and heterogeneous equivalence checking. As a result, the paper currently feels more like a well-executed benchmark integration paper with useful empirical results than a clearly standout ICML benchmark contribution. That does not make it weak overall, but it does hurt its chances in a venue where benchmark papers are expected to set a very high standard on rigor, novelty of evaluation framing, and long-term field impact.

---

> ### Author Rebuttal · Authors · 2026-03-28
>
> Thank you for the careful and highly professional review. We agree the key issues are benchmark validity, interpretation clarity, and positioning. We respond briefly below.
>
> 1. **Mixed equivalence regime.**
>
> We agree this is the most important clarification. In CIRBench, `Equiv` is a layered best-available semantic guard, not a single uniform proof notion. For verifier-clean candidates, we use Alive2 whenever it returns a definite result on single-function cases; checksum fallback is used only when Alive2 is timeout / unknown / unsupported / OOM, or when the case is module-level and Alive2 is outside scope. We never override an Alive2 `not-eq` verdict with harness agreement.
>
> **Table 1. Equivalence-checking regime breakdown**
>
> | Subset       | Granularity | % Alive2 definite | % checksum fallback | Fallback reason                       |
> | ------------ | ----------- | ----------------- | ------------------- | ------------------------------------- |
> | Repair       | function    | 100%              | -                   | -                                     |
> | Refactor     | function    | 100%              | -                   | -                                     |
> | Transform    | function    | 71.25%            | 28.75%              | timeout / unknown / unsupported / OOM |
> | Module-level | module      | -                 | 100%                | not applicable                        |
>
> We will also place Alive2-only results next to the main `Equiv@k` numbers.
>
> **Table 2. Single-function Transform: overall Equiv vs. Alive2-only Equiv**
> *(`E@k` and `Ef@k` use the same single-function denominator; `E@k` counts equivalence established by either Alive2 or checksum fallback, while `Ef@k` counts only equivalence established by Alive2.)*
>
> | Model    | E@1  | Ef@1 | E@5  | Ef@5 |
> | -------- | ---- | ---- | ---- | ---- |
> | GPT-5    | 76.7 | 51.0 | 94.2 | 60.4 |
> | Claude   | 35.0 | 15.1 | 62.5 | 27.1 |
> | Gemini   | 38.3 | 36.0 | 56.7 | 48.5 |
> | Grok     | 33.8 | 22.4 | 77.9 | 49.0 |
> | Qwen     | 43.3 | 26.6 | 81.7 | 52.6 |
> | DeepSeek | 19.2 | 16.1 | 50.0 | 36.0 |
>
> The main conclusion is unchanged: current models are much stronger at producing verifier-clean IR than at reliable semantics-preserving IR rewriting.
>
> 2. **Table 5 and main-text clarity.**
>
> We agree Table 5 is not self-explanatory enough, and too many qualifications are currently appendix-only. In the revision, we will move into the main text:
>  (i) the mixed equivalence regime;
>  (ii) the interpretation of Table 5 (counting unit, denominator, overlap rule);
>  (iii) the contamination/provenance disclaimer;
>  (iv) the positioning against prior work.
>
> 3. **Contamination / memorization.**
>
> We agree CIRBench should not be read as contamination-free. Our claim is narrower: CIRBench evaluates reliability at the LLVM IR boundary under verifier / equivalence / performance constraints. Instances come from public suites, but are not evaluated as raw source programs; they are compiled to LLVM IR and then undergo slicing, normalization, task-specific construction, deduplication, UB filtering, and timing filtering. We view this as reducing trivial source-level transfer, while not ruling out overlap. We will make this limitation explicit and add a compact provenance table.
>
> 4. **Positioning vs. prior work.**
>
> We agree the distinction from prior IR-level work should be sharper. CIRBench is not only “more tasks”: it places Analysis / Repair / Refactor / Transform at the same LLVM IR interface under one correctness-aware protocol combining verifier validity, semantic equivalence, and runtime/performance. This unified setup is what lets us expose the central gap in the paper: models can appear plausible or even verifier-clean at IR level without being reliably semantics-preserving.
>
> **Table 3. Comparison to prior IR-level resources and evaluations**
>
> | Work                    | IR   | A    | Rp   | Rf   | T    | V    | E    | P    |
> | ----------------------- | ---- | ---- | ---- | ---- | ---- | ---- | ---- | ---- |
> | SLTrans / IRCoder       | ✓    | ✗    | ✗    | ✗    | ✗    | ✗    | ✗    | ✗    |
> | ProGraML / DeepDataFlow | ✓    | ✓    | ✗    | ✗    | ✗    | ✗    | ✗    | ✗    |
> | IR-OptSet               | ✓    | ✓    | ✗    | ✗    | ✓    | ◐    | ◐    | ✓    |
> | LLM-Vectorizer          | ◐    | ✗    | ✗    | ✗    | ✓    | ✗    | ✓    | ✓    |
> | CIRBench                | ✓    | ✓    | ✓    | ✓    | ✓    | ✓    | ✓    | ✓    |
>
> 5. **Interpretation limits.**
>
> We agree mixed checking regimes constrain comparability across subsets, especially between single-function and module-level cases. We will state this explicitly and report them separately.
>
> 6. **Limitations / impact.**
>
> We agree the current discussion is too generic. We will make it more concrete by clarifying that verifier-passing outputs are not equivalent to semantic safety, benchmark success is not deployment readiness, and conclusions may not transfer cleanly across LLVM version drift / heterogeneous IR dialects.

---

> > ### Author Rebuttal · Reviewer_5hqD · 2026-04-04
> >
> > Thank you for the clarification and additional details. I understand the authors’ point better now and appreciate the response.

---

### Official Review · Reviewer_uFT4 · 2026-03-13

**Soundness:** 4
**Presentation:** 4
**Significance:** 4
**Originality:** 4
**Overall Recommendation:** 5
**Confidence:** 4

**Summary:**

The authors present CIRBench, a benchmarking set for seeing how well LLMs perform four tasks on LLVM IR. A total of 800 snipped span Analysis (100), Repair (200), Refactoring (200) and Optimisation (300) and are separated into easy/medium/hard categories. An evaluation harness is provided that automatically evaluates results using SOTA tools and methods for IR verification.

The authors then evaluate a few LLMs on this harness achieving mixed results, showing that this benchmark has not yet been "solved" and remains an interesting axis on which models can be evaluated.

**Compliance With Llm Reviewing Policy:**

Affirmed.

**Final Justification:**

I keep my recommendation to accept since I believe the authors to have written a strong paper

**Key Questions For Authors:**

All good

**Limitations:**

Yes

**Strengths And Weaknesses:**

# Strengths

The paper presents a well thought out set of tasks including a evaluation harness for evaluation. This represents a significant contribution to the field, with implications both for the ML and the compiler community.

# Weaknesses

The appendix is very large, but mainly consists of high-quality content. It would have been neat to have more of this present in the main paper, but then I am unsure what could have been cut. I guess this really would benefit from a higher page limit.

---

> ### Author Rebuttal · Authors · 2026-03-25
>
> Thank you for the positive assessment and for recognizing CIRBench as a meaningful benchmark for LLVM IR reasoning and rewriting. We are encouraged that you view the benchmark as not yet “solved,” which is exactly the behavior we hoped to obtain from a discriminative IR-level evaluation.
>
> We also appreciate your suggestion on the balance between the main paper and the appendix. In the revision, we will move the most essential implementation details and prompt / I/O specifications from the appendix into the main text to improve readability and self-containment.

---

> > ### Author Rebuttal · Reviewer_uFT4 · 2026-04-01
> >
> > Thank you!

---

### Decision · Program_Chairs · 2026-04-30

**Decision:**

Accept (spotlight)

**Comment:**

This paper proposed a comprehensive benchmark designed to evaluate the capability of LLMs in terms of the reasoning and manipulation capability of LLVM IR. It focuses on "compiler-grade" tasks across four distinct tracks: analysis, repair, refactor and transformation. There are 800 instances across different levels of difficulties, and compared to -O3 as the baseline, the authors found interesting aspects of advantages, as well as areas to improve in the current LLM’s capability.

Overall reviewers are all generally satisfied with the significance and the practical implications of this benchmark, furthermore the benchmark and toolchain are open-sourced, which makes it immediately usable and can be a high-impact resource for the research community.

Although there can be some concerns like contamination, and some more clarity is needed in the main paper, overall we believe this paper made a solid contribution to the field and can be a solid benchmark for future research, if this benchmark can keep evolving as well.